# Doubled strength and ductility via maraging effect and dynamic precipitate transformation in ultrastrong medium-entropy alloy

Hyun Chung[1,7], Won Seok Choi[2,3,7], Hosun Jun [2], Hyeon-Seok Do[4], Byeong-Joo Lee [4], Pyuck-Pa Choi [2], Heung Nam Han [5], Won-Seok Ko[6] & Seok Su Sohn [1]✉

Demands for ultrahigh strength in structural materials have been steadily increasing in response to environmental issues. Maraging alloys offer a high tensile strength and fracture toughness through a reduction of lattice defects and formation of intermetallic precipitates. The semi-coherent precipitates are crucial for exhibiting ultrahigh strength; however, they still result in limited work hardening and uniform ductility. Here, we demonstrate a strategy involving deformable semi-coherent precipitates and their dynamic phase transformation based on a narrow stability gap between two kinds of ordered phases. In a model medium-entropy alloy, the matrix precipitate acts as a dislocation barrier and also dislocation glide media; the grain-boundary precipitate further contributes to a significant work-hardening via dynamic precipitate transformation into the type of matrix precipitate. This combination results in a twofold enhancement of strength and uniform ductility, thus suggesting a promising alloy design concept for enhanced mechanical properties in developing various ultrastrong metallic materials.

Martensite is a very hard microconstituent in structural materials formed by shear or displacive transformation, originally found in quenched steels by German scientist Martens[1]. The technological importance of martensite primarily comes from its high strength based on hierarchy substructures, while in most carbon steels the martensitic microstructures are subjected to tempering that assigns greater toughness by increasing ductility but decreasing strength. As a special class of very low-carbon steels hindering formations of brittle carbides, maraging alloys (martensite+ageing) achieve a desirable combination of strength and toughness while maintaining relatively high ductility. The maraging can restore the ductility with the reduction of lattice defects formed during martensitic transformation and exploit an additional hardening effect through the formation of nanosized intermetallic precipitates[2–7], instead of various carbides as in carbon-bearing tempered martensite. However, the introduced large coherency strains with heterogeneous distribution of semi-coherent precipitates may lead to crack initiation as a double-edged sword[8–10]. Besides, the uniform ductility is limited to ~2% due to the limited work hardening in commercial maraging alloys[6,7,11] exhibiting a yield strength of 2 gigapascals (GPa) or higher, thus requiring further enhancement of both strength and work hardening for their widespread applications.

As a feasible strategy for overcoming the limited work hardening, we can take advantage of dynamic phase transformation, known as transformation-induced plasticity (TRIP)[12–16]. The transformation from a relatively soft parent phase into the hard martensite under

[1]Department of Materials Science and Engineering, Korea University, Seoul 02841, South Korea. [2]Department of Materials Science and Engineering, Korea Advanced Institute of Science and Technology, Daejeon 34141, South Korea. [3]Institute of Environmental Science and Technology, SK Innovation, Daejeon 34124, South Korea. [4]Department of Materials Science and Engineering, Pohang University of Science and Technology, Pohang 37673, South Korea. [5]Department of Materials Science and Engineering, Seoul National University, Seoul 08826, South Korea. [6]Department of Materials Science and Engineering, Inha University, Incheon 22212, South Korea. [7]These authors contributed equally: Hyun Chung, Won Seok Choi. ✉e-mail: sssohn@korea.ac.kr

mechanical deformation results in a high work-hardening and postponing necking phenomenon. Thus, the TRIP effect has been intensively studied to obtain tough structural materials[17–19]. The dynamic phase transformation is enabled by the knowledge-based modification of chemical composition to narrow down the phase stability gap between the parent phase and the resulting martensite. Then, the mechanical loading and deformation can initiate the martensitic transformation at a service temperature. Nevertheless, the limitation of this design concept is the inherently soft parent phase, typically disordered face-centred cubic (fcc) phase. This phase starts to plastically deform at a lower stress level compared to the hard martensite, eventually resulting in a low yield strength[20–23].

Here, we demonstrate a strategy utilising versatile semi-coherent precipitates as effective dislocation obstacles and also pursuing dynamic precipitate transformation to significantly improve both yield strength and work-hardening behaviour in an initially hard martensitic $FeCo_{0.8}V_{0.2}$ medium-entropy alloy (MEA) as a model alloy. MEAs, as a subclass of alloys termed high-entropy alloys (HEAs), multi-principal element alloys, or compositionally complex alloys, consist of generally 3–4 elements at high concentrations, where the high configuration entropy supports the formation of solid-solution phase rather than intermetallic compounds[24]. Those alloys exhibit remarkable mechanical properties which originate from high solid-solution strengthening or severe lattice distortion due to large differences in atomic volumes and electronegativity of constituent elements[25,26]. Based on this strong matrix, precipitates occurring dynamic transformation are selected through Ab initio calculations that the 50Co–25Fe–25 V precipitates show an indistinct difference in the phase stability between hP24 ($Al_3Pu$-type) (ordered hexagonal close-packed (hcp) structure) and $L1_2$ (ordered fcc structure). The narrow stability gap eventually leads to a flexible structure selection depending on the nucleation site; hP24 locate within matrix and $L1_2$ at grain boundary. The hP24 within the matrix primarily acts as dislocation obstacles and the metastable $L1_2$ at the grain boundary contributes to a high work-hardening rate by dynamic precipitate transformation into hP24. This combination drives notable mechanical performances, resulting in a twofold enhancement of both strength (up to 2.1 GPa) and uniform ductility (about 4.0%). The results provide a promising alloy design concept aiming at multiple semi-coherent precipitates with a narrow stability gap to encourage the flexible precipitation behaviour and dynamic precipitate transformation for ultrastrong and tough structural materials.

## Results
### Alloy design
In the model Fe-rich Fe–Co–V ternary system, the first element (Fe) has an initiating role in forming a high-strength matrix of body-centred cubic (bcc) martensite, whereas the second (Co) and third (V) elements

assist in the formation of various intermetallic compounds as strengthening phases during ageing treatment, e.g. the $Co_3V_1$ type hP24 compound observed in the Co–V binary system[27]. For the current alloy system, density functional theory (DFT) calculations predicted the relative stability of possible close-packed ordered structures with hexagonal symmetry (hP24 and $D0_{19}$) and with cubic symmetry ($L1_2$) compounds in varying compositions of $(Fe,Co)_3V_1$ (see details of computational methodology and additional data in Methods and Supplementary Fig. 1).

The formation energies at 0 K (Fig. 1a) indicate that the hP24 and $L1_2$ compounds are more stable than the $D0_{19}$ compound. Furthermore, the highest stability of the hP24 compound was obtained when the composition was $Co_3V_1$, which is consistent with reported binary phase diagrams[27]. Besides, the relative stability between the hP24 and $L1_2$ compounds changes with a larger Fe concentration, exhibiting an almost identical stability ($\Delta E$ ~1 meV/atom at 0 K) at approximately the $Co_2Fe_1V_1$ composition. DFT calculations at finite temperatures (Fig. 1b) further confirmed a negligible difference in phase stability even at higher temperatures, implying the feasibility of dynamic phase transformation under mechanical responses.

Moreover, we confirmed that the $Co_2Fe_1V_1$ compounds are expected to form at a certain stage of the heat treatment process (e.g. quenching + ageing), when they are nucleated from the bcc matrix phase. The driving force for the formation of hP24, which is defined by the free energy difference between the parent phase (bcc) and hP24, i.e. $\triangle G = G_{hP24} - G_{bcc}$ in Fig. 1c, is largest in $Co_2Fe_1V_1$ compounds especially at typical temperatures for the ageing treatment. Then, we performed thermodynamic calculations to form the desired precipitates in the Fe–Co–V ternary system (see details in Methods and Supplementary Figs. 2 and 3). Thus, $FeCo_{0.8}V_{0.2}$ was selected in order to obtain fully martensitic microstructure after quenching as the matrix and form precipitates (PPTs) of the desired $M_3V$ phase (M: Fe, Co) after ageing without other phases causing embrittlement. To fabricate the maraging MEA with dynamic precipitate transformation, the cold-rolled alloy was solution-annealed at 1173 K (900 °C) for 10 min (hereafter, referred to as SA) and subsequently aged at 823 K (550 °C) for 1 h and 24 h (hereafter, referred to as 1H and 24H, respectively).

### Microstructure and precipitation behaviour
Figure 2a exhibits X-ray diffraction (XRD) patterns that only bcc peaks were identifiable in the SA and 1H alloy, while fcc and hcp phases were present in addition to the bcc matrix for 24H alloy. Figure 2b shows scanning electron microscopy (SEM) image and electron-backscatter diffraction (EBSD) phase map of the SA alloy, revealing a single bcc phase with a fine grain size of $3.1 \pm 2.1 \, \mu m$. Although only weak substructures with a fine-grained structure were identified in the SA alloy (Fig. 2b), prolonged solution treatment at 1273 K (1000 °C) for 1 h and

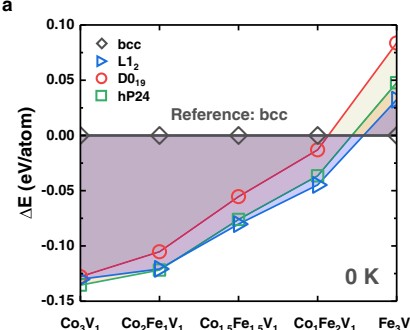

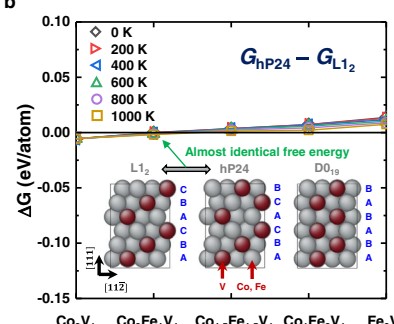

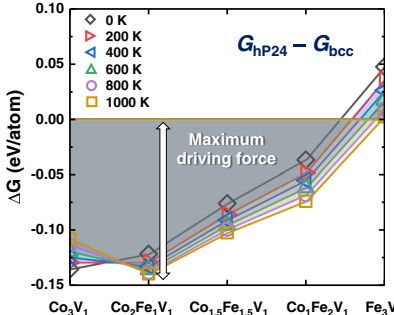

**Fig. 1 | First-principles density functional theory (DFT) calculation of phase stability in $A_3B_1$-type (A: (Fe,Co), B: V). a** Stability of candidate ordered precipitates (hP24, $L1_2$, and $D0_{19}$) with respect to disordered body-centred cubic (bcc) solid solution at 0 K predicted via DFT calculation. Temperature dependence of Gibbs free energy (G) difference between (**b**) hP24 and $L1_2$, and (**c**) hP24 and disordered bcc solid solution, approximated by Debye–Grüneisen model. Schematics of configurations are shown in the inset. Connecting lines between symbols are only for visual guidance.

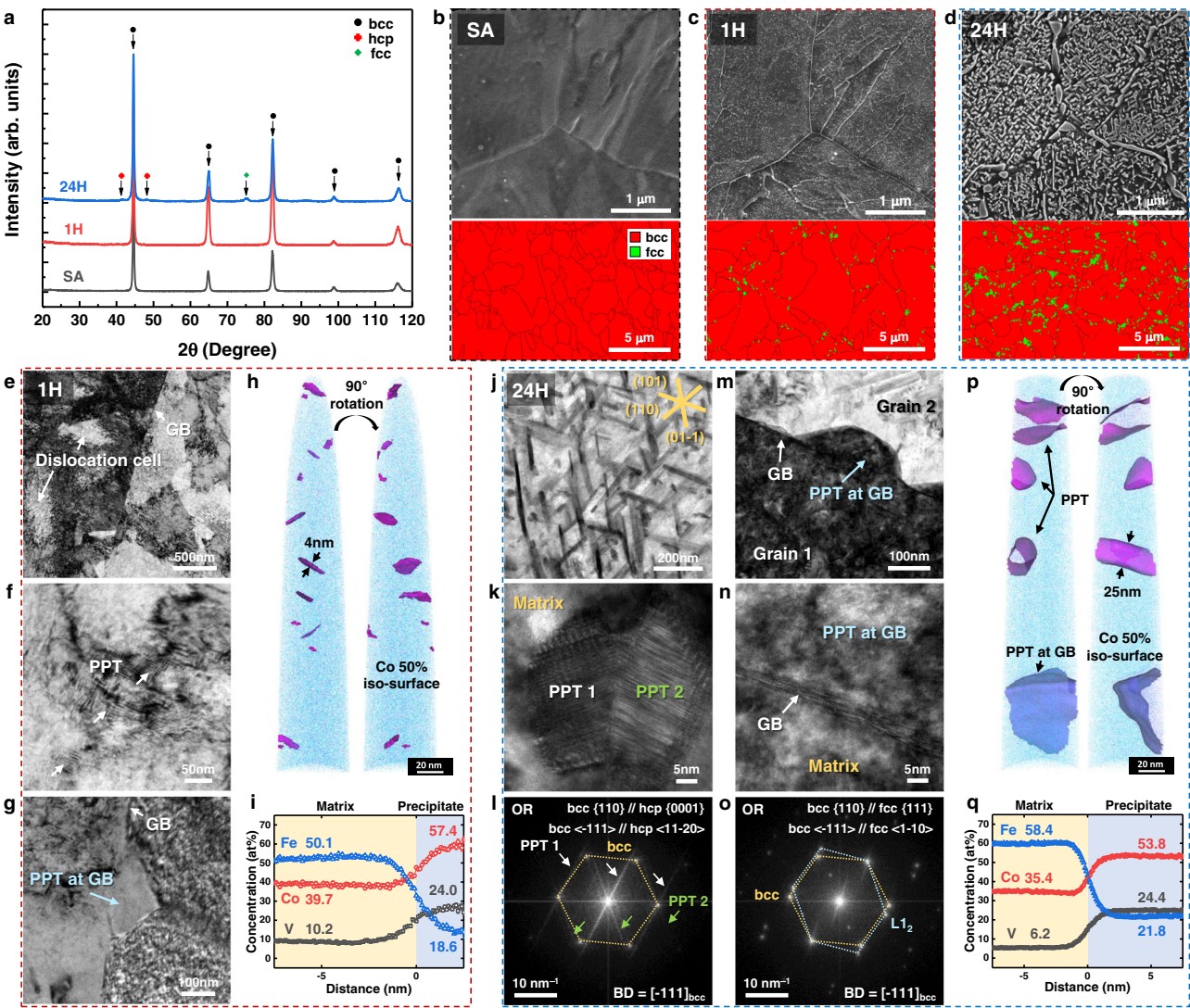

**Fig. 2 | Characterisation of precipitates upon ageing conditions. a** Phase identification via X-ray diffraction (XRD) analysis for SA, 1H, and 24H alloys. **b–d** Scanning electron microscopy (SEM) images and electron-backscatter diffraction (EBSD) phase maps for different ageing times, (**e–i**) transmission electron microscopy (TEM) images and atom probe tomography (APT) reconstruction and proximity histogram across precipitates and bcc matrix for 1H alloy. The 50 at% Co iso-concentration surface shows the reference phase boundary. **j,k,m,n** TEM images, (**l,o**) corresponding fast Fourier-transform (FFT) images, (**p,q**) APT reconstruction and proximity histogram for 24H alloy.

consequent coarse-grained prior fcc phase ensure the characteristic martensitic structure, including packet, block, and lath substructures (Supplementary Fig. 4). No evident secondary phases were observed in the SA alloy. On the other hand, SEM images of the 1H alloy (Fig. 2c) reveal fine PPTs within the matrix grains and film-like PPTs decorating prior fcc grain boundaries (PFGBs). Although all of those PPTs were hardly identifiable in the EBSD phase map, the PPTs at the PFGBs were confirmed to have an fcc-based structure. Upon further ageing for 24 h (24H in Fig. 2d), PPTs within the matrix developed rod shapes and became interwoven to each other, whereas the films at the PFGBs grew into fine polygonal particulates with an increased area fraction.

The crystal structures of the PPTs were further characterised via transmission electron microscopy (TEM), with images shown in Fig. 2e–g for 1H and Fig. 2j–o for 24H. Their compositions were identified via atom probe tomography (APT), with images shown in Fig. 2h,i for 1H and Fig. 2p,q for 24H. The TEM results confirmed that two kinds of PPTs form in the bcc matrix (Fig. 2f,j) and at the grain boundaries (Fig. 2g,m) during ageing, and that the size and volume fraction of the PPTs increase with ageing time. In the 1H alloy, the shapes of the PPTs in the bcc matrix are not fully defined (Fig. 2f), whereas in the 24H

alloy, their shapes become rod-like on the {110} planes as triangular-shaped clusters decorated with dense stacking faults (SFs) (Fig. 2j,k). An APT reconstruction of the 1H alloy clearly reveals the very small size (width of ~4 nm) of the PPTs (Fig. 2h), whereas those in 24H alloy are determined to exhibit average length and diameter of ~160 nm and ~25 nm, respectively (Fig. 2p). The local lattice structure was confirmed to be a hP24 (Al$_3$Pu-type) structure (ordered hcp), but most of the fast Fourier-transform (FFT) images do not show clear diffraction patterns because of the high density of SFs (Fig. 2k,l). For large-sized PPTs, high-resolution TEM (HRTEM) and FFT images (Fig. 2l and Supplementary Fig. 5) confirm the orientation relationship (OR) between hP24 with the dense SFs and bcc matrix to be {110}$_{bcc}$// {0001}$_{hcp}$, <$\bar{1}$11>$_{bcc}$//<11$\bar{2}$0>$_{hcp}$, i.e. Burgers OR, whereas they have a diffused phase interface because of the SFs. These semi-coherent interfaces enable the PPT to maintain the nanometre size after further ageing up to 1 week (Supplementary Fig. 6). Unlike the grain interior hP24, the PPTs along the PFGB are defect-free (Fig. 2n) and exhibit a clear L1$_2$ (Cu$_3$Au-type) diffraction pattern (Fig. 2o). Observation of the interface between L1$_2$ and the bcc matrix identifies an OR of {110}$_{bcc}$//{111}$_{fcc}$, <$\bar{1}$11>$_{bcc}$//<1$\bar{1}$0>$_{fcc}$, i.e. Kurdjumov–Sachs (K–S) OR. A

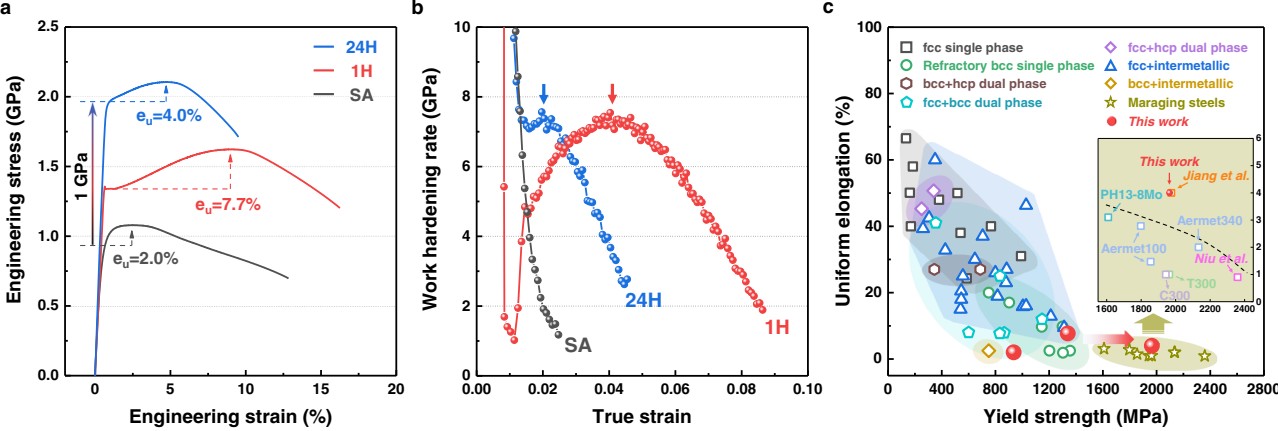

**Fig. 3 | Room-temperature mechanical properties of the alloys. a** Engineering tensile stress–strain curves of SA (black), 1H (red), and 24H (blue) alloys (e$_u$: uniform elongation). **b** Work-hardening rate versus true strain for SA, 1H, and 24H alloys. **c** Comparison of yield strength versus uniform elongation for the FeCo$_{0.8}$V$_{0.2}$ MEAs and other single or multiphase high-/medium-entropy alloys and maraging steels.

proximity histogram analysis across these phases for 24H alloy reveals that there is no difference in the chemical compositions of both PPTs despite the microstructural differences observed in the TEM results (Supplementary Table 1). Based on the measured chemical compositions of the bcc matrix (58.4Fe–35.4Co–6.2 V, atomic percent) and PPTs (23.7Fe–51.7Co–24.6 V, atomic percent), the volume fraction of the PPTs, including both hP24 and L1$_2$, was calculated to be ~25.5% using the lever rule[5,28]. Similarly, the volume fraction of PPTs in the 1H alloy was estimated to be ~3.9% based on the chemical compositions of the bcc matrix (50.1Fe–39.7Co–10.2 V, atomic percent) and PPTs (18.6Fe–57.4Co–24.0 V, atomic percent).

Our DFT calculations show a narrow stability gap between the hP24 and L1$_2$ phases (Fig. 1). It predicted that both hP24 and L1$_2$ have the lowest Gibbs free energy and the highest probability to precipitate when the composition is approximately Co$_2$Fe$_1$V$_1$, which well corresponds to experimental findings. Notably, PPTs possess two different crystal structures depending on the nucleation sites. The interior hP24 PPTs develop Burgers OR with the matrix, while the grain-boundary L1$_2$ PPTs develop K-S OR with the matrix. Based on the ORs of each PPT, the measured lattice misfits exhibited a value of 0.46% for hP24 and 1.84% for L1$_2$. PPTs having low-energy interfaces, e.g. Burgers and K-S ORs, exhibit flat interfaces leading to rod-shaped morphology for hP24 and polygonal shape for L1$_2$ and both interfaces are parallel to the {011}$_{bcc}$. It has been well-established that a PPT at grain boundary has a rational OR, e.g. K-S, with an adjacent grain, while the interface is incoherent with the other adjacent grain and highly dependent on the grain boundary characteristics[29]. However, it was also shown that PPTs at grain boundary become partially coherent by formations of ledges and misfits compensating defects[30], as they try to reduce the increment of interfacial energy at most. It was also noted that the activation energy increases with increasing the tilt angle between the low-energy interface and the original matrix grain boundary, when the tilt angle is below the critical value[31]. In other words, the closest {111}$_{fcc}$ to the matrix grain boundary (~{011}$_{bcc}$) is selected as a low-energy interface[32]. Therefore, both the interior PPTs with a highly faulted structure and the grain-boundary PPTs showing K-S OR with an adjacent grain are expected to minimise the interfacial energy. While it is confirmed that L1$_2$ develops K-S OR with an adjacent grain (Fig. 2o), the interface structure with the other adjacent grain does not show exact K-S OR. As expected, the irrational OR was observed where the beam directions (BDs) were BD$_{bcc}$ = <111> and BD$_{L12}$ ~ <114 >, respectively; however, the HRTEM image (Supplementary Fig. 7) shows an interesting contrast from the phase interface with partially coherent bonding and its arrangement changes with the curved interface. This

might be related to the formation of ledges and the misfit compensating defects to minimise the interfacial energy[30].

To further explain the origin of the precipitation behaviour of the two phases, a selection of structure based on electron concentrations (e/a) of PPTs and their heterogeneous nucleation and growth were further considered. According to the theory based on e/a from Liu et al. [33], the M$_3$V phase begins to show hexagonality mixed with a cubic crystal structure over e/a of 7.89. The e/a value exhibits ~7.81 for our actual composition of the present PPTs, which is lower than the critical concentration forming hexagonality. Therefore, it is likely for the PPTs to form cubic ordered structure, i.e. L1$_2$. However, in the process of heterogeneous nucleation and growth of PPTs, those nucleating on dislocations are dominated by the strain-field effect where the lattice misfit becomes a critical factor[34]. The interior PPTs would accommodate numerous stacking faults and transform in the direction of the hP24 structure during growth to minimise the misfit (L1$_2$–bcc: 1.84% versus hP24–bcc: 0.46%). The narrow energy stability gap between two phases also seems to allow the formation of SFs and local hP24 structure. On the other hand, the PPTs at grain boundaries consume the PFGBs and forms the semi-coherent interface with an adjacent grain to lower the interface energy. This well corresponds to a conventional heterogeneous precipitation mechanism at grain boundaries. When L1$_2$ forms a semi-coherent interface with an adjacent grain, it seems that the formation of SFs does not further reduce the energy, but the locally flat interfaces form with the other adjacent grain, resulting in the polygonal-shaped PPT aforementioned.

## Mechanical properties

Figure 3a presents the room-temperature engineering stress–strain curves of the SA, 1H, and 24H alloys (see summarised details in Supplementary Table 2). The SA alloy exhibited limited work hardening with a yield strength of 956 ± 20 MPa and short uniform elongation of 2.0 ± 0.1%, which is a well-known behaviour of traditional martensitic steels[4,35]. By comparison, ageing for 1 h resulted in an increase in the yield strength by 393 MPa and enhanced the work-hardening capacity with a uniform elongation of 7.7 ± 0.5%. Work-hardening rate plots (Fig. 3b) confirm that the SA alloy demonstrates a gradual decrease, whereas the 1H alloy exhibits a notable increase to ~7.5 GPa. Of particular interest is the effect of 24 h ageing. The yield strength was more than doubled to 1965 ± 24 MPa, and larger uniform elongation of 4.0 ± 0.1% was obtained even at approximately 2 GPa strength, which are twofold enhancements compared to the corresponding properties in the SA alloy. The tensile strength reached 2105 ± 17 MPa with the high work-hardening capability, which has not been reported for

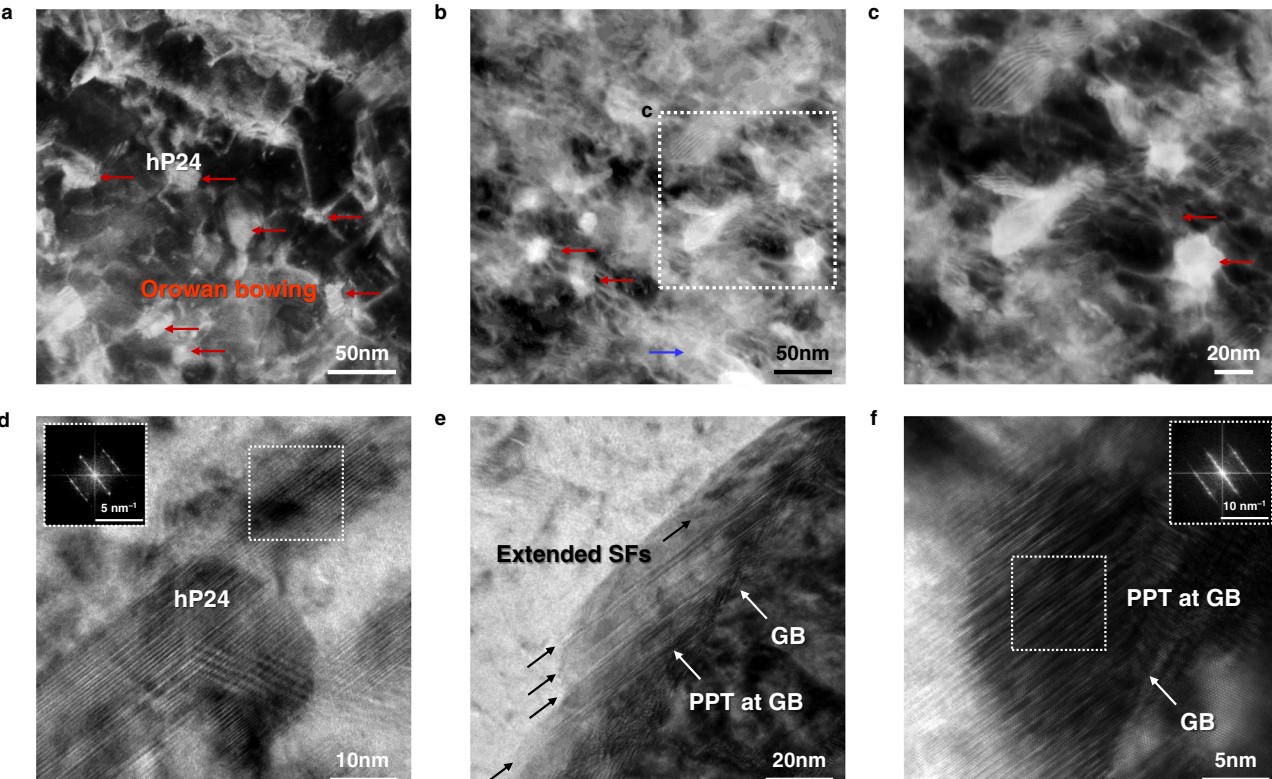

**Fig. 4 | Deformation mechanisms of 24H alloy. a–f** Transmission electron microscopy (TEM) images of deformed microstructure. **a** Dislocations bowing when bypassing the cross-section of hP24 marked by red arrows. **b** Massive dislocation interactions and homogeneous deformation substructures indicated by a blue arrow. **c** An enlarged image of the marked area in (**b**) showing hP24 acting as obstacles. **d** Highly faulted hP24 precipitates with more evident hcp diffraction pattern of hP24 structure. **e** Partial dislocations emitted from grain boundaries (GBs) leading to extended stacking faults (SFs) in L1$_2$. **f** L1$_2$ transforming to hP24 with the aid of deformation-induced SFs.

existing ultrahigh-strength maraging alloys with semi-coherent PPTs[36–38]. These properties are compared in Fig. 3c with those of other solid-solution or multiphase HEAs and MEAs, and maraging steels (detailed data sources of the different alloys are provided in Supplementary References). The levels of performance of the SA and 1H alloys are comparable mainly to those of refractory bcc and multiphase bcc alloys, whereas the 24H alloy exhibits an exceptional combination of ultrahigh strength and uniform ductility, surpassing those previously reported for ultrastrong precipitation-strengthened HEAs, MEAs, and maraging steels. The inset in Fig. 3c is the magnified region of ultrahigh strength maraging group to clearly distinguish the properties. When it comes to ultrahigh strength metallic materials with strength reaching near 2 GPa, most of them show uniform ductility of <2%, while the present 24H alloy reaches 4%.

## Discussion

To elucidate the strengthening and ductilisation mechanisms by maraging effects, the deformed microstructures (tensile strained by 1%) of 24H alloy were investigated through TEM analyses as shown in Fig. 4. It is likely that fine PPTs significantly enhance the yield strength; either by shearing or Orowan bowing mechanism. The dark-field image in Fig. 4a shows the top view of the rod-shaped PPTs, where the direct observation of dislocation-precipitate interactions indicates that gliding dislocations bypass by the Orowan bowing mechanism. The strength increment from the precipitation strengthening of interior hP24 is estimated to be ~1030 MPa (see Methods and Supplementary Fig. 8 for detailed information). On the other hand, the strengthening contribution from forest dislocations decreases from 493 MPa to 377 MPa as the alloy undergoes ageing for 24 h. The reduction in dislocation density is attributed to the combination of thermal recovery

(dislocation annihilation and rearrangement) and consumption by forming PPTs with semi-coherent interfaces. As the ageing proceeds for 1 h and 24 h, the dislocation density of the alloy gradually decreases from $1.43 \times 10^{15}$ m$^{-2}$ in SA to $9.49 \times 10^{14}$ m$^{-2}$ in 1H and $8.40 \times 10^{14}$ m$^{-2}$ in 24H. These two different contributions have a counter effect; however, the large increase in strength due to precipitation renders the decrease in strength due to the reduction in dislocations relatively negligible. However, this precipitation strengthening, on its own, cannot be a unique mechanism for the notable performance of the alloys examined in this study. The aged 24H alloy exhibits greater ductility, specifically uniform elongation, despite its ultrahigh strength level.

The increased strength and ductility of the present alloy are attributed to the following three dominant mechanisms: (1) dislocation behaviours in the matrix; (2) SF formation in interior PPT (hP24); and (3) TRIP effect in grain-boundary PPT (L1$_2$). First, Fig. 4b,c shows the deformed substructure of the matrix and that adjacent to the interfaces between the interior PPTs and matrix. High-density dislocations form homogeneous deformation substructures as indicated by a blue arrow, and the high fraction and small interspacing of PPTs lead to massive dislocation interactions in the matrix. Before ageing, the as-quenched SA alloy initially possesses high dislocation density due to inherent characteristics of martensitic transformations. This initial high-density tangled dislocation has limited capability of work hardening as represented in Fig. 3a,b. However, the reduction in dislocation density due the ageing treatment (24H alloy) increases the mean free path of dislocations (MFP). This increased MFP allows uniform dislocation glides at a certain regular spacing (see Fig. 4b,c) and consequent high ductility. However, it cannot be concluded that this mechanism is solely dominant in strengthening and ductilisation due to massive interior PPTs with the average interparticle spacing of

~58 nm, which limits the substantial increase of MFP. Nevertheless, the uniform dislocation glides and homogeneous deformation substructures contribute to preventing premature cracking in ultrahigh-strength alloys as observed in Supplementary Fig. 9.

Secondly, turning the focus to the PPTs, it is observed that the interior hP24 PPTs show the glide of SFs within them as well as the formation of Orowan loops with matrix dislocations. Although the interior PPTs after ageing already contain dense SFs (see Fig. 2k and Supplementary Fig. 5), the FFT image of deformed PPTs shown in Fig. 4d reveals a more evident pattern of the hP24 structure. This result indicates that partial dislocations in the PPTs can glide by applied stress, leading to the well-defined highly faulted structure. Therefore, it is concluded that the interior PPTs accompany dislocation glides, which contributes to plasticity as well as the precipitate strengthening via the Orowan mechanism.

Thirdly, whereas the interior PPTs (hP24) possess dense SFs prior to deformation, the grain-boundary PPTs (L1$_2$) remain as defect-free states (Fig. 2n). Plastic deformation introduces partial dislocations motion and SFs formation within the L1$_2$ that result in the dynamic phase transformation into hP24, leading to considerable work-hardening and large uniform ductility. It is confirmed that the TRIP occurs after yielding (Fig. 4e,f), and the high critical stress for TRIP can be estimated by an increasing high work hardening rate after yielding in contrast to the SA sample (Fig. 3b)[39–41]. Figure 4e,f clearly shows the gradual progress of deformation and resulting phase transformation during deformation. Partial dislocations were emitted from the grain boundary, resulting in the extended SFs and dynamic precipitate transformation into hP24. The presence of grain-boundary phases allowing to accommodate plastic deformation by phase transformations would prevent premature exhaustion of dislocation sources at the boundaries through the repeated generation of dislocation.

The current dynamic precipitate transformation at boundaries has similar effects to TRIP effects in Mn steels or quenching and partitioning steels that consist of metastable austenite at the boundaries[41,42]. It is well known that the dynamic phase transformation of the metastable phase in TRIP steels and alloys postpones plastic instability and thus enhances ductility and work hardening in a large scope[43,44]. The high fraction of metastable austenite contributes to considerable plasticity; however, it also accompanies a decrease in yield strength, and thus this kind of microstructure cannot implement a class of ultrahigh strength steel. In order to maintain the ultrahigh strength level and gain additional ductility, a fraction of the soft phase should keep a minimum, or its morphology should be tuned to possess high strength level and high mechanical stability. As for the represented case, Aermet100 contains 1–6% metastable austenite, which presents as thin foils with no downside to the strength. Although it is hard to exhibit a significant increment of work-hardening rate or ductility in a large scope such as general high-strength TRIP steels, it can contribute to preventing premature failure and improving toughness effectively, as observed in several studies of Aermet100, PH13-8 Mo, and Mn steels[40,45,46]. Therefore, it is worth mentioning that the TRIP effect in ultrahigh strength alloys exhibits different performances from the conventional high-strength TRIP steels. In this respect, the present work exploits inherently hard intermetallic phases, which also have no downside to the strength, and their deformable and transformable characteristics provide a twofold enhancement in strength and ductility via ageing.

The underlying mechanism of dynamic structural changes from L1$_2$ to hP24 can be understood based on the stacking faults pair. As observed in Ni$_3$(Al,Ti) precipitation-hardened nickel-based alloys, L1$_2$ shearing by matrix dislocations can result in different structures of SnNi$_3$-type DO$_{19}$, TiNi$_3$-type DO$_{24}$, and VCo$_3$-type hP24[47]. The stacking sequence of L1$_2$ is ABCABCA…, whereas that of hP24 is ABCACBA…, which exhibits twin-like formation. The stacking sequence of L1$_2$ can be changed to hP24

through the shear displacements of the type {111}1/3 <112> with superlattice intrinsic (S-ISF) and extrinsic (S-ESF) stacking faults pair[47]. To further illustrate the sequence changes, a schematic drawing of the layers is provided (Supplementary Fig. 10), where the modification of sequences via SFs is shown. Through <112>-type shear displacement of partial dislocations, S-ESF adds a C´ layer between $A_2$ and $B_2$, whereas S-ISF removes C$_2$ layer from the sequence. As a result, the extrinsic/intrinsic stacking faults pair leads to a sequence from A$_1$B$_1$C$_1$A$_2$B$_2$C$_2$ to A$_1$B$_1$C$_1$A$_2$C´B$_2$_A$_3$ as ABCACBA…, which is that of hP24, i.e. the VCo$_3$ type.

In this respect, the glide of partial dislocations enables L1$_2$ to hP24 precipitate transformation during deformation, introducing additional phase boundaries and SFs. The introduced interfaces effectively reduce the mean free path of dislocations and even hinder the activation of secondary slip systems due to the limited number of slip systems in the hcp structure[13,48]. This mechanism, therefore, significantly contributes to work hardening through the dynamic Hall–Petch effect. The phase transformation also relieves the strain energy accumulated during tensile deformation, enabling further plastic deformation to be accommodated. Furthermore, the cubic to hexagonal transformation in the A$_3$B-type structure is known to significantly increase frictional stress due to the interplanar-locking effects[33,49]. Therefore, unlike the conventional ordered PPTs or carbides that contribute to only strengthening, PPTs in the present alloy are able to implement both strengthening and ductilitisation mechanisms. Notably, this dynamic phase transformation and consequent ductilisation of ultrastrong alloys are attributed to the narrow stability gap of the desired multiple PPTs. The deformation mechanisms of the PPTs were sketched in Fig. 5 with microstructural evolutions during ageing.

In summary, we demonstrate a design strategy resulting in an ultrahigh strength of ~2 GPa and acceptable uniform elongation of ~4.0% through deformable hP24 and transformable L1$_2$ PPTs. Controlling the relative stability of ordered phases with compositional variation in (Fe,Co)$_3$V assigns the metastability, enabling dynamic phase transformation of intermetallic compounds. The soft matrix or disordered second phase in conventional TRIP alloys leads to a relatively low yield strength compared to those of martensitic alloys. Thus, the present initial hard martensite matrix and semi-coherent intermetallic compounds result in no downside of yield strength for ensuring the ultrastrong metallic materials. As well as the strength, the present alloy overcomes limited work-hardening and ductilisation behaviour by adopting the SF formation in interior PPTs and the TRIP effect in grain-boundary PPTs. These complex metallurgical mechanisms with simple heat treatment are suggested to implement high-performance and load-bearing application requirements. We expect these multiple semi-coherent precipitates and the dynamic precipitate transformation to be applicable to the development of next-generation ultrastrong metallic materials.

## Methods
### DFT calculation
We performed first-principles DFT calculations using Vienna ab initio Simulation Package code[50–52] and the projector augmented wave method[53] within a generalised-gradient approximation of Perdew–Burke–Ernzerhof[54] for the exchange-correlation functional. For V and Fe, we used pseudopotentials where the semi core $p$ states are regarded as part of the valence. A cut-off energy of 400 eV was used for the plane wave basis set, and the Methfessel–Paxton smearing method was applied with a width of 0.1 eV. $\Gamma$-centred $k$-point meshes with a density of ~9000 $k$-points per reciprocal atom were employed for all calculations. Magnetism was included with spin-polarised calculations for all supercells taken into account. Atomic positions were relaxed using a conjugate gradient algorithm with the convergence criteria for energy and forces set to $10^{-6}$ eV and $10^{-2}$ eV Å$^{-1}$, respectively. To take into account the possible disordering of atoms in each

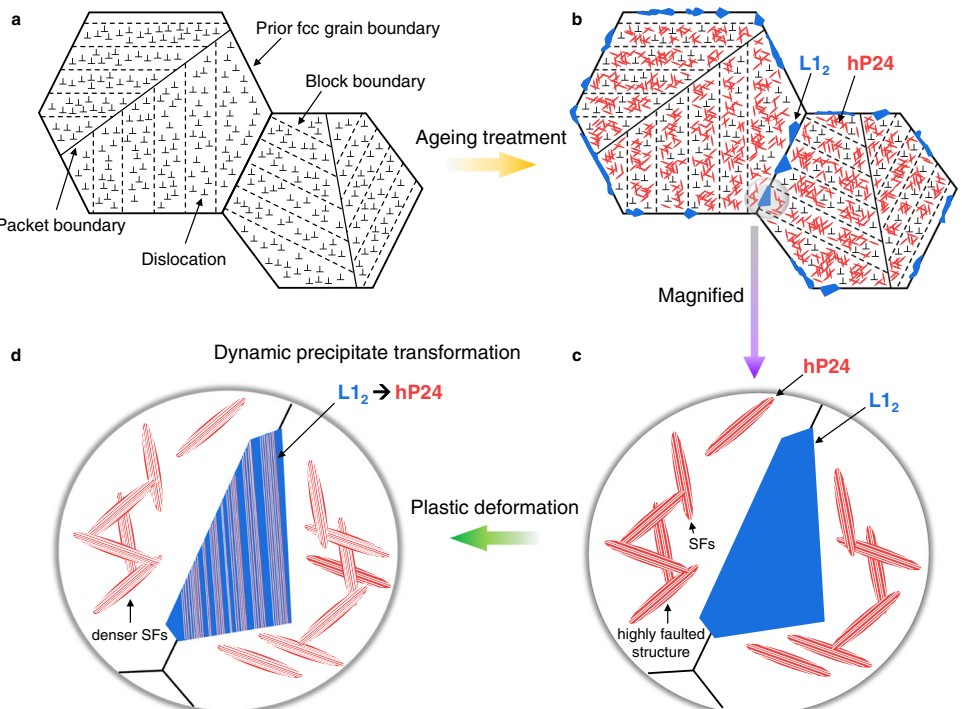

**Fig. 5 | Schematic drawings illustrating the microstructure evolutions.**
**a** Martensitic microstructure with hierarchical substructures and high dislocation density. **b** Massive precipitation of hP24 within matrix and L1$_2$ along grain boundaries after ageing treatment. **c** Magnified illustration showing highly faulted structure of rod-like hP24 and defect-free L1$_2$ with flat interfaces and (**d**) evolved precipitates upon plastic deformation, hP24 possess denser SFs and L1$_2$ is also introduced with SFs leading to local dynamic precipitate transformation to hP24.

structure, special quasi-random structures (SQS) were constructed based on a Monte Carlo algorithm (MCSQS code) from the Alloy Theoretic Automated Toolkit[55,56]. The relative stabilities of structures at finite temperature were examined with inclusion of the vibrational contribution to the free energy. The vibrational entropy was estimated based on the Debye–Grüneisen model according to a methodology presented in a previous study[57].

### Alloy design & fabrications

The FeCo$_{0.8}$V$_{0.2}$ alloy was designed based on the possible precipitation of M$_3$V-type close-packed ordered compounds with different stacking sequences and varying Fe and Co compositions. Earlier investigation roughly predicted the stabilities of those compounds based on a concept of electron concentration and an assumption that Fe and Co atoms occupy the same sublattice (i.e. M(Fe,Co)$_3$V)[33]. Our DFT calculations predicted that such site occupation would indeed be the most feasible way for stabilising such close-packed compounds, and that the relative phase stabilities of compounds can be changed diversely through variations in the Fe and Co composition (Fig. 1a).

As aforementioned, three elements of Fe, Co, and V are the candidates for our strategy. To determine the alloy composition to embody the desired precipitates, we performed thermodynamic calculations based on CALPHAD approaches using Thermo-Calc software with a TCFE2000 database and its upgraded version[58–61]. The phase diagrams under fixed V content at 10 at% and 20 at% are shown in Supplementary Fig. 2. For the Fe$_x$Co$_{80-x}$V$_{20}$ phase diagram, the sigma (σ) phase is present along the Fe-rich regions, which is likely to cause embrittlement. As mentioned in the manuscript, however, sufficient Fe is necessary in order to obtain martensitic microstructure. Thus, the V content of 10 at% was considered to avoid brittle σ phase, and also to obtain aimed M$_3$V phase in the martensitic matrix. At 10% V, in determining the proportion of Fe and Co, calculation results in Supplementary Fig. 3 demonstrate that more abundant Co leads to a massive fraction of M$_3$V. Interestingly, the M$_3$V phase has the composition

aimed Co$_2$Fe$_1$V$_1$ in the Fe$_{50}$Co$_{40}$V$_{10}$ alloy composition. Further increase of nominal Co content leads to more Co content (>50 at%) in the M$_3$V phase, which deviates from the targeted Co$_2$Fe$_1$V$_1$. Thus, considering the calculation results and the possibility of the formation of brittle sigma phase in the abundant V composition, we selected Fe$_{50}$Co$_{40}$V$_{10}$ as a bulk alloy composition with relatively higher Fe and Co content compared to that of V. This composition is expected to obtain the aimed Co$_2$Fe$_1$V$_1$ precipitate for M$_3$V precipitate and also to form fully martensitic structures from single fcc phase at a high-temperature range (>900 °C).

Moreover, with regard to nomenclature, FeCo$_{0.8}$V$_{0.2}$ can be referred to as a MEA based on the classification by Yeh et al. [62], where the criteria can be expressed with respect to the ideal configurational mixing entropy, which is 5.76–11.52 J mol$^{-1}$ K$^{-1}$ for MEAs. Our FeCo$_{0.8}$V$_{0.2}$ alloy has an ideal configurational mixing entropy of 7.84 J mol$^{-1}$ K$^{-1}$ based on the bulk composition, which fits into the criteria for MEAs. Ingots with a nominal composition of FeCo$_{0.8}$V$_{0.2}$ were fabricated via vacuum induction melting (model: MC100V, Indutherm GmbH, Germany) using high-purity pure elements (Fe-99.9%, Co-99.95%, V-99.95%), zirconia crucible, and graphite mould (100 × 35 × 8 mm$^3$). The ingots were homogenised at 1373 K (1100 °C) for 6 h under an Ar atmosphere, followed by water quenching. The surface scale was removed via pickling in a 20% HCl solution for 20 s, after which the ingots were cold rolled to a ~80% thickness reduction. The cold-rolled sheets, now down to a thickness of 1.5 mm, were annealed at 1173 K (900 °C) for 10 min to produce fully recrystallised fine fcc grains and subsequently quenched to form the martensitic matrix. Ageing treatments were conducted at 823 K (550 °C) for 1 h and 24 h under an Ar atmosphere and followed by water quenching.

### Mechanical tests

Flat dog-bone specimens were prepared with a gauge length, gauge width, and thickness of 12, 4 and 1.5 mm, respectively. The tensile specimens were cut from the annealed or aged sheets via electrical

discharge machining. Uniaxial tensile tests were conducted using a universal testing machine (model: 8801, Instron, Canton, MA, USA) at a strain rate of $1 \times 10^{-3} \, s^{-1}$.

## Microstructural characterisation

The evolved crystal structure was identified via X-ray diffraction (XRD, X'Pert PRO-MRD, PHILIPS, Netherlands). The grain structure and phase distribution were investigated via EBSD using field-emission scanning electron microscopy (FE-SEM, S-4300SE, HITACHI, Japan). Specimens were mechanically polished using SiC papers of up to 4000 grit size, and electropolished in a mixed solution of 92% acetic acid and 8% perchloric acid. Transmission electron microscopy (TEM) was performed using a JEOL JEM-2100F instrument operated at 200 kV. The TEM specimens were prepared via focused ion beam lift-out using a FEI Helios NanoLab 450 F1 instrument. The chemical composition of each phase was measured via atom probe tomography (APT, Cameca LEAP 4000X HR) using the pulsed laser mode at a specimen base temperature of -50 K. The pulse frequency and energy were 200 kHz and 50 pJ, respectively. The acquired APT data were reconstructed and analysed using the commercial IVAS® software by Cameca. To reveal the deformation structures, TEM was conducted for the 1%-deformed tensile specimen.

## Determination of lattice parameter and misfit

Lattice parameters used for determining lattice misfit were obtained from the XRD analysis; 2.866 Å for the bcc matrix; 3.576 Å for the L1$_2$; 4.942 Å and 11.957 Å for the $a$ and $c$ of hP24, respectively. The lattice misfits between the precipitates and the matrix were estimated by using the equation $\delta = 2(d_{PPT} - d_{matrix})/(d_{PPT} + d_{matrix})$, where $d_{PPT}$ and $d_{matrix}$ are the lattice constant of hP24 or L1$_2$ and bcc martensite. The K-S OR is based on the cubic-cubic relationship, while Burgers OR is based on a hexagonal-cubic relationship. Therefore, $d_{hP24}$ is considered as lattice constant along $a$ axis of hP24 and $d_{bcc}$ as $\sqrt{3}$-fold of the lattice constant of bcc martensite.

## Estimation of strengthening contributions

An approximation of the yield strength can be drawn through a summation of four individual contributions as follows:

$$\sigma_{ys} = \sigma_0 + \triangle\sigma_{gb} + \triangle\sigma_\rho + \triangle\sigma_{ppt} \tag{1}$$

where $\sigma_0$, $\triangle\sigma_{gb}$, $\triangle\sigma_{ppt}$, and $\triangle\sigma_{ppt}$ are solid-solution strengthening, grain-boundary strengthening, dislocation strengthening, and precipitation strengthening, respectively.

The matrix of the SA alloy contains 10 at% of V and the 24H alloy contains 6 at% V. There might be a slight difference in the absolute values for solid-solution strengthening stress; however, we focused on the changing and competing contributions from dislocation strengthening and precipitation strengthening. Hence, we assumed that the solid-solution strengthening stress of 360 MPa and the Hall–Petch coefficient of 220 MPa μm$^{1/2}$ from the previous investigation[63]. The high-angle boundaries are considered for grain size calculation.

The contribution of dislocation strengthening was quantified from the well-known Taylor hardening equation (Eq. (2))[64]:

$$\triangle\sigma_\rho = M\alpha Gb\rho^{1/2} \tag{2}$$

where $M = 2.733$ is the Taylor factor for the bcc crystal, $\alpha = 0.25$ is a constant for bcc alloys, $G$ is the shear modulus[65], and $b$ is the Burger's vector. The dislocation density ($\rho$) was obtained from the XRD data using modified Williamson-Hall method[66] with contrast factors and elastic anisotropy[67,68]. Peak positions and line broadening (represented by the full width at half maximum, FWHM) data of five representative peaks of bcc, i.e. (110), (200), (211), (220), and (310), were extracted to

estimate dislocation densities of the matrix. We plotted $\Delta K$ and $KC^{1/2}$ from the five representative bcc peaks data and fitted linear slope, quantitatively analysing the line broadening. The equation used for fitting the slope is as follows:

$$\triangle K \cong \frac{0.9}{D} + \left(\frac{\pi\kappa^2 b^2}{2}\right)^{\frac{1}{2}} \rho^{\frac{1}{2}} K C^{\frac{1}{2}} + O(K^2 C) \tag{3}$$

where $\theta$ is the diffraction angle, $\lambda$ is the wavelength of the X-rays, $K$ is $2\sin\theta/\lambda$, $\Delta K$ is $(2\Delta(2\theta)\cos\theta/\lambda)$, $D$ characterises the crystallite size, $b$ is the Burgers vector, $C$ is the average contrast factor of dislocations, and $\kappa$ is a constant depending on the effective outer cut-off radius of the dislocations. $O$ stands for the higher-order terms where $O(K^2 C)$ is considered negligible.

Two types of precipitates are present in the current alloy where only the intragranular precipitate (hP24) can have a significant effect on strengthening. The hP24 precipitates have a width diameter of ~25 nm for 24H alloy. Considering the size of precipitates and their semi-coherent interface, it is difficult to shear them by dislocation glide as observed in Fig. 4a. Therefore, Orowan bowing mechanism is regarded for the precipitation strengthening which is expressed as follow[69]:

$$\Delta\sigma_{ppt} = M \frac{0.4Gb}{\pi(1 - v)^{1/2}} \frac{\ln(\frac{2\bar{r}}{b})}{\lambda} \tag{4}$$

where $v$ is the Poisson ratio, $\bar{r}$ is the average radius of the precipitates, and $\lambda$ is the average interspacing between precipitates. Equation (4) is used for ideally spherical precipitates, but the present hP24 exhibits rod-shaped morphology where the height is longer than the diameter ($h = c/a > 1$). Thus, taking account of the extended morphology and variation of interparticle spacing, we used the modified equation as Eq. (5) from Sonderegger et al. [70]. The additional consideration of how the aspect ratio affects the precipitation strengthening is expressed in Eqs. (5) and (6):

$$K = h^{\frac{1}{6}} \left(\frac{2 + h^2}{3}\right)^{-\frac{1}{4}}, \tag{5}$$

$$\Delta\sigma_{ppt-modified} = K^{-1}\Delta\sigma_{ppt} \tag{6}$$

where $K$ is a shape correction factor, $h$ is the aspect ratio of the particle. The hP24 has an average aspect ratio of 6.4 for 24H alloy.

## Data availability

The data that support the findings of this study are available from the corresponding author upon request.

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

## Acknowledgements

This work was supported by the National Research Foundation of Korea grant (NRF – 2020R1C1C1003554); the NRF of Korea grant funded by the Korea government (MSIT) (NRF-2022R1A5A1030054); the Fundamental Research Program of the Korean Institute of Materials Science (PNK8730); Samsung Research Funding & Incubation Center of Samsung Electronics (SRFC-MA1902-04); and the Korea Institute for Advancement of Technology (KIAT) grant funded by the Korean Government (MOTIE, P0002019, The Competency Development Program for Industry Specialist). H.N.H. was supported by the NRF of Korea grant funded by the Korea government (MSIT) (NRF – 2020R1A5A6017701). W.-S.K. was supported by the NRF of Korea funded by Ministry of Science and ICT (Grant No. NRF-2019M3D1A1079214) and INHA UNIVERSITY Research Grant.

## Author contributions

S.S.S. designed the research. H.C. fabricated materials and conducted mechanical tests and SEM-EBSD observations. W.S.C., H.J. and P.-P.C conducted TEM and APT experiments. W.-S.K. performed ab initio calculations. H.-S.D. and B.-J.L. performed thermodynamic calculations. H.C., W.S.C., H.N.H., W.-S.K. and S.S.S. analysed the data and wrote the paper. All authors reviewed and contributed to the final paper.

## Competing interests

The authors declare no competing interests.
