## [Peer Review File · Nature Communications]

Doubled strength and ductility via maraging effect and dynamic precipitate transformation in ultrastrong medium-entropy alloyREVIEWER COMMENTS

Reviewer #1 (Remarks to the Author):

This paper reports ultrahigh strength and good ductility in a tempered FeCo-based MEA owing to both metastability nature of the introduced hard precipitates at grain boundaries and the staking-fault-induced plasticity of relatively stable grain interior precipitates. TRIP effect has been frequently reported in ultrahigh strength steels (e.g., Aermet100) in which the thin metastable foil austenite has no downside to the strength. Nevertheless, strengthening by deformable hard precipitates was not reported before and may have certain importance for further development of ultrahigh strength materials. Overall, the methodology needs to improve substantially and some issues raised below must also be addressed.

1. The strength increment after the applied ageing is very high, exceeding 1000 MPa. The authors have estimated strengthening from different contributors. However, the models applied were not clearly elaborated. Since the paper reported that the precipitates can be deformed by SFs or by TRIP effect, which should be described by neither cutting nor Orowan looping. Is the strengthening response related to the critical stress and/or strain to initiate SF or TWIP? The authors showed that their energy is nearly identical, which means that one of them is highly unstable?
2. It is well known that the stability of microstructure is also critical for cracking resistance of ultrahigh strength alloys. The authors characterized evolution of the nanoprecipitates after deformation in detail, but the information regarding deformed substructure of the matrix and that adjacent to the interfaces between the nanoprecipitates and matrix were not provided. While these substructures are definitely important for the understanding of both the strengthening and ductilization mechanism, which may also separate the contributions from interior precipitation and from grain boundary precipitates. The discussion in the current version is too simple, and both strengthening and ductilization mechanisms should be discussed more systematically.
3. In page 6, the authors defined the driving force for the formation of hP24, i.e., the free energy difference between the parent phase (bcc) and hP24 structure. The composition used for the calculation is the stoichiometric composition of the involved precipitates phase, rather than the alloy composition. As such, the DFT calculation might be able to explain origins of the TRIP effect of the metastable phase, but cannot reveal any direct relationship with the designed alloy compositions. That is, from alloy design point of view, these calculations are insufficient to provide any hints.
4. The authors pointed out that boundary precipitates is L12 rather than hp 24 because of the difference in both lattice misfit and variant amount. First, the close-packed plane for L12 and hp24 should be identical and they are both the parallel plane in K-S and Burgers OR. Therefore, I guess the misfit of the L12-matrix and hp24-matrix is very similar. Also, the authors should provide the basic data such as lattice parameter, used for the misfit calculation. Second, Figure 2 shows that boundary precipitates mainly grow into one side of grain boundary with obvious orientation relationship. According to the description that they maintain semi-coherent interfaces with both side matrix, the orientation relationship with the other side of grain should be also provided.
5. This paper mainly focused on semi-coherent precipitates. Given the large size, is the interface still semi-coherent?
6. No scale bar in all IFFT and SAED patterns. Without it, a pattern can be identified to be many phases.
7. Figure 3c, data for recently developed maraging steels should be included. It would be nice to discuss possible difference of underlying mechanisms involved.

Reviewer #2 (Remarks to the Author):

This article presents work assessing the maraging effect in a FeCo_{0.8}V_{0.2} alloy. It comprises some interesting analysis and is a good demonstration of a joint modelling & experiment approach. Unfortunately, I'm not convinced of the significance of the phase transformation mechanisms proposed, nor that the alloy developed has particularly special mechanical properties. I have the following specific comments (in order of appearance in the text):

1. Introduction text: "preliminary" should be primarily?
2. To say that the technological importance of martensite originates from mar-aging is an odd statement. Most martensitic microstructures in use today (i.e., those in steels) are not age hardened, but rather just tempered (softened). It is tempering that allows martensite to be useful in most cases. The statement seems to contradict the statement later in the paragraph, which says that current maraging alloys aren't applied much (indicating that maraging isn't very technologically important).
3. More explanation is needed when the calculations on 50Co-25Fe-25V are introduced. The reader has no idea what the expected microstructure of FeCo_{0.8}V_{0.2} should be, and therefore why calculations of Co₂FeV are important.
4. There's no such thing as an fcc L12 phase. L12 is primitive.
5. A uniform ductility of 4% doesn't seem very impressive. Should it be seen as so?
6. The technological importance of FeCo_{0.8}V_{0.2} is never discussed. Why chose this alloy? Why these elements?
7. The alloy design section only very briefly addressed the possibility of transformation between the ordered phases – e.g., it didn't state which transformation was going to be targeted. Also, it's not clear why such a transformation should be beneficial. In TRIP steels, there is a dramatic increase in ductility owing to the considerable shape and volume change associated with the formation of martensite (so you get a lot of hardening and a lot of strain). What about in this case?
8. In Fig. 2a, there are features that do not look like clean bcc grains. Is this the "substructure" that is mentioned? What is meant by "substructure"?
9. It is stated that "the proximity histogram across the PPT and bcc matrix (Fig. 2h) implies that precipitation involving atomic diffusion was not yet completed." What aspect implies this is the case? The smooth transition in composition? And what is the reasoning?
10. It is suggested that the tendency of one precipitate to nucleate at grain boundaries compared to the other can be related to the number of variants in the OR. This is an unusual suggestion – can the authors provide evidence of where this has been demonstrated before? It still seems unlikely that good correspondence would be found on both sides of the boundary very often, even when there are lots of possible variants.
11. I feel Fig. 3 should include some conventional maraging alloys, since many at least meet the properties found in this work (e.g., Aermet 340).
12. More discussion is needed on the mechanism proposed for the work hardening – presumably, the precipitate phase transformation leads to a harder precipitate than originally? How does this mean that more plastic deformation can be accommodated after the transformation (which is the next sentence)?
13. As sketched in Fig. 5, the phase transformation plays no role in the grain interiors – surely this is where work hardening is most important? Therefore, is the phase transformation effect of limited impact?
14. Can the authors comment on whether the precipitates in conventional high-strength maraging alloying also exhibit similar deformation characteristics – e.g., high densities of SFs?

Reviewer #3 (Remarks to the Author):

This manuscript is mainly focusing on the design and development of $\text{Co}_{0.8}\text{Fe}_{0.2}\text{V}$ medium-entropy alloy with enhanced strength, work-hardening, and ductility, which are dominantly contributed by the formation of semi-coherent precipitates and dynamic precipitate transformation.

The Co-Fe-V alloy was well designed by DFT calculation and thoroughly studied by advanced experiment approaches, such as TEM and APT.

Overall, I felt that the authors did nice work, and the manuscript is well written. The contents of the manuscript are enough to attract significant attention in HEA society, since the design strategy of precipitate-strengthened MEA/HEA is still not well established, compared to its solid-solution-strengthened MEA/HEA. Thus, I would like to suggest this work be considered for publication after minor revision.

I have some minor comments/questions on the manuscript as shown below.

1. The authors categorized the studied $\text{Co}_{0.8}\text{Fe}_{0.2}\text{V}$ alloy as a medium entropy alloy. However, there is no description of the uniqueness of M/HEAs. The M/HEA cannot be only defined by a number of alloy components and their atomic ratio. Please include the definition/description of M/HEA and its benefits to properties (such as entropy effect, lattice distortion, and etc.) in the introduction section.

2. Is there any specific reason why the authors have designed this alloy by DFT rather than CALPHAD? If so, please include the advantage of DFT approach for alloy design compared to CALPHAD, since most of HEAs have been designed by either empirical rules and CALPHAD.

3. How the authors obtained the dislocation densities for 3 conditions heat-treated alloys?

4. The XRD data should move to the main text.

5. The current status of the paper should be re-organized. For instance, some parts of the strengthening and plasticity mechanisms section should be the discussion part and there is no scientific discussion in the current discussion part. The current discussion part looks like a summary part to me.

6. The target applications for developed M/HEAs are mostly high-temperature. Have authors tried to investigate the mechanical behavior at elevated temperature for the present alloy?

Dear Reviewers,

We would like to submit the following revised manuscript for publication in *Nature Communications*, on behalf of all co-authors.

Title: Doubled strength and ductility via maraging effect and dynamic precipitate transformation in ultrastrong medium-entropy alloy

Authors: Hyun Chung, Won Seok Choi, Hosun Jun, Hyeon-Seok Do, Byeong-Joo Lee, Pyuck-Pa Choi, Heung Nam Han, Won-Seok Ko, Seok Su Sohn*

We cordially thank the reviewers for considering our work and providing helpful and valuable comments. We carefully revised the manuscript accordingly, and a detailed point-by-point reply to each item is enclosed below. Revised or added sentences have been highlighted in blue in the revised manuscript. Please kindly consider the attached reply and revised manuscript for further consideration.

- Response to the Reviewer 1: Pages 2 – 26
- Response to the Reviewer 2: Pages 27 – 51
- Response to the Reviewer 3: Pages 52 – 63

Sincerely yours,

Seok Su Sohn

Associate Professor, Korea University

E-mail. sssohn@korea.ac.kr

Reviewer #1 (Remarks to the Author):

This paper reports ultrahigh strength and good ductility in a tempered FeCo-based MEA owing to both metastability nature of the introduced hard precipitates at grain boundaries and the staking-fault-induced plasticity of relatively stable grain interior precipitates. TRIP effect has been frequently reported in ultrahigh strength steels (e.g., Aermet100) in which the thin metastable foil austenite has no downside to the strength. Nevertheless, strengthening by deformable hard precipitates was not reported before and may have certain importance for further development of ultrahigh strength materials. Overall, the methodology needs to improve substantially and some issues raised below must also be addressed.

Question #1-1> The strength increment after the applied ageing is very high, exceeding 1000 MPa. The authors have estimated strengthening from different contributors. However, the models applied were not clearly elaborated. Since the paper reported that the precipitates can be deformed by SFs or by TRIP effect, which should be described by neither cutting nor Orowan looping. Is the strengthening response related to the critical stress and/or strain to initiate SF or TWIP? The authors showed that their energy is nearly identical, which means that one of them is highly unstable?

Reply #1-1> We appreciate the reviewer's helpful comment, allowing us to improve the explanation of strengthening mechanisms more clearly. We had regarded the hP24 precipitates as non-shearable ones due to their relatively large size (average length and diameter: ~160 nm and ~25 nm) for the 24H alloy. According to the reviewer's suggestion, we have performed additional TEM analyses to directly unravel the dislocation behaviours as shown in the revised Fig. 4a. This dark-field image shows the top view of the rod-shaped precipitates indicated by red arrows. Direct observations of the dislocation-precipitate interaction indicate that gliding dislocations bypass by Orowan bowing mechanisms. The contribution of this strengthening

was estimated to be ~ 1030 MPa as shown in Supplementary Fig. 8, and thus it is reasonable to conclude that the yield strength increment after ageing results from the precipitation strengthening of hP24 in the matrix. The following figures and figure captions have been revised in Fig. 4.

Figure 4. Deformation mechanisms of 24H alloy. a–f Transmission electron microscopy (TEM) images of deformed microstructure. **a** Dislocations bowing when bypassing the cross-section of hP24 marked by red arrows. **b** Massive dislocation interactions and homogeneous deformation substructures indicated by a blue arrow. **c** hP24 acting as obstacles. **d** Highly faulted hP24 precipitates with more evident hcp diffraction pattern of hP24 structure. **e** Partial dislocations emitted from grain boundaries (GBs) leading to extended stacking faults (SFs) in L12. **f** L12 transforming to hP24 with the aid of deformation-induced SFs.

As the reviewer pointed out, the formations of SF and metastability of L12 are crucial mechanisms in this work. However, they are more likely associated with work-hardening mechanisms and consequent plasticity, rather than directly determining the increment of yield strength. We observed that the interior hP24 precipitates show the glide of SFs within them

during deformation as well as the formation of Orowan loops with matrix dislocations. Although the precipitates after ageing already contain dense SFs (see Fig. 2k and Supplementary Fig. 5), the FFT image of deformed precipitates shown in Fig. 4d reveals more evident pattern of the hP24 structure. This result indicates that partial dislocations in the precipitates can glide by applied stress, leading to the L1₂ to hP24 transformation. Therefore, it is concluded that the interior precipitates accompany dislocation glides, which contributes to plasticity as well as the precipitate strengthening via the Orowan mechanism.

As well as the formation of SFs in interior hP24 precipitates, the grain-boundary L1₂ precipitates also significantly contribute to work hardening by TRIP effect. It is expected that the contribution of L1₂ to the yield strength increment is relatively negligible because dislocations hardly interact with the grain-boundary precipitates before yielding in contrast to the grain interior precipitates. They can pile up at the L1₂ interfaces (*i.e.*, the location near grain boundaries) in the later stage of deformation. Thus, partial dislocations motion and SF formation within the L1₂ precipitates result in the dynamic phase transformation into hP24, leading to a considerable work-hardening and large uniform ductility.

Moreover, the grain-boundary L1₂ precipitate is thought to be metastable rather than highly unstable. If the L1₂ precipitates are highly unstable, they are prone to phase transformation under stress levels before yielding (known as stress-induced transformation), which can hardly contribute to plasticity. It was confirmed that TRIP occurs after yielding (Fig. 4e,f), and the high critical stress for TRIP can be estimated by an increasing high work hardening rate after yielding in contrast to the SA sample (Fig. 3b). Fig. 4e,f clearly shows the gradual progress of deformation and resulting phase transformation during deformation: partial dislocations were emitted from the grain boundary, resulting in the extended SFs and precipitate transformation into hP24. Therefore, it can be concluded that the mechanical stability of the grain-boundary

L1₂ precipitate is enough to contribute to the high work hardening and prolonged plastic deformation. Therefore, the following sentences and relevant references have been added to support the explanation of strengthening mechanisms in the discussion part.

Discussion

- ... shearing or Orowan bowing mechanism. The dark-field image in Fig. 4a shows the top view of the rod-shaped PPTs, where the direct observation of dislocation-precipitate interactions indicates that gliding dislocations bypass by the Orowan bowing mechanism. The strength increment from the precipitation strengthening of interior hP24 is estimated to be ~1030 MPa (see Methods and Supplementary Fig. 8 for detailed information). On the other hand, the strengthening contribution...
- ... it is observed that the interior hP24 PPTs show the glide of SFs within them as well as the formation of Orowan loops with matrix dislocations. Although the interior PPTs after ageing already contain dense SFs (see Fig. 2k and Supplementary Fig. 5), the FFT image of deformed PPTs shown in Fig. 4d reveals a more evident pattern of the hP24 structure. This result indicates that partial dislocations in the PPTs can glide by applied stress, leading to the well-defined highly faulted structure. Therefore, it is concluded that the interior PPTs accompany dislocation glides, which contributes to plasticity as well as the precipitate strengthening via the Orowan mechanism.
- ... Plastic deformation introduces partial dislocations motion and SFs formation within the L1₂ that result in the dynamic phase transformation into hP24, leading to considerable work-hardening and large uniform ductility. It is confirmed that the TRIP occurs after yielding (Fig. 4e,f), and the high critical stress for TRIP can be estimated by an increasing high work hardening rate after yielding in contrast to the SA sample (Fig. 3b)³⁹⁻⁴¹. Figure 4e,f clearly shows the gradual progress of deformation and resulting phase transformation during deformation. Partial dislocations were emitted

from the grain boundary, resulting in the extended SFs and dynamic precipitate transformation into hP24. The presence of grain-boundary phases allowing to accommodate plastic deformation by phase transformations would prevent premature exhaustion of dislocation sources at the boundaries through the repeated generation of dislocation.

References

- [39.] Pierce, D. T., Jiménez, J. A., Bentley, J., Raabe, D. & Wittig, J. E. The influence of stacking fault energy on the microstructural and strain-hardening evolution of Fe-Mn-Al-Si steels during tensile deformation. *Acta Mater.* **100**, 178–190 (2015).
- [40.] Schnitzer, R. *et al.* Influence of reverted austenite on static and dynamic mechanical properties of a PH 13-8 Mo maraging steel. *Mater. Sci. Eng. A* **527**, 2065–2070 (2010).
- [41.] Gao, G. *et al.* Enhanced ductility and toughness in an ultrahigh-strength Mn-Si-Cr-C steel: The great potential of ultrafine filmy retained austenite. *Acta Mater.* **76**, 425–433 (2014).

Question #1-2> It is well known that the stability of microstructure is also critical for cracking resistance of ultrahigh strength alloys. The authors characterized evolution of the nanoprecipitates after deformation in detail, but the information regarding deformed substructure of the matrix and that adjacent to the interfaces between the nanoprecipitates and matrix were not provided. While these substructures are definitely important for the understanding of both the strengthening and ductilization mechanism, which may also separate the contributions from interior precipitation and from grain boundary precipitates. The

discussion in the current version is too simple, and both strengthening and ductilization mechanisms should be discussed more systematically.

Reply #1-2> We appreciate the reviewer for providing constructive suggestions to unravel the deformation mechanism concerning the matrix. According to the reviewer's suggestions, we have conducted more TEM analyses as shown in the revised Fig. 4b,c (please see the figure set in **Reply #1-1**). Fig. 4b,c shows the deformed substructure of the matrix and that adjacent to the interfaces between the interior hP24 precipitates and matrix. High-density dislocations form homogeneous deformation substructures as indicated by a blue arrow, and the high fraction and small interspacing of hP24 precipitates lead to massive dislocation interactions in the matrix. The hP24 effectively acts as obstacles to dislocation motions. To further understand the strengthening and ductilisation mechanism, details are discussed as follows.

The significant strengthening and ductilisation of the present alloy are attributed to the following three dominant mechanisms: 1) dislocation behaviours in the matrix; 2) SF formation in interior hP24; 3) TRIP effect in grain-boundary L1₂. First, aforementioned, high-density dislocations generate homogeneous deformation substructures, leading to massive interfacial interactions with hP24 obstacles. Before ageing, the as-quenched SA alloy possesses high dislocation density initially due to inherent characteristics of martensitic transformations. This initial high-density tangled dislocation has limited capability of work hardening as represented in Fig. 3a,b. However, the ageing treatment (24H alloy) enables thermal recovery (dislocation annihilation and rearrangement) and increases the mean-free path of dislocations (MFP). In addition to the recovery, dislocations are consumed by formations of semi-coherent interfaces of hP24 and L1₂ as measured from $1.43 \times 10^{15} \text{ m}^{-2}$ to $8.40 \times 10^{14} \text{ m}^{-2}$ via XRD. This increased MFP allows uniform dislocation glides at a certain regular spacing (Fig. 4b,c) and consequent high ductility. However, it cannot be concluded that this mechanism is solely dominant due to

massive interior precipitates with the average interparticle spacing of ~58 nm, which limits the substantial increase of MFP. Nevertheless, the uniform dislocation glides and homogeneous deformation substructures prevent premature cracking in ultrahigh strength alloys.

In this regard, the deformation mechanisms of hP24 and L1₂, described in **Reply #1-1**, contribute to the enhanced work hardening and delayed plastic instability, resulting in both strengthening and ductilisation. It is reasonable that deformable hP24 and transformable L1₂ precipitates play an essential role in enhancing ductilisation, resulting in the doubled uniform elongation to 4% with the tensile strength of ~2100 MPa. However, the quantitative contributions of described three mechanisms are challenging to estimate due to their complex and interdependent effects. The following sentences have been added to discuss the strengthening and ductilisation mechanisms in the discussion part.

- **Discussion.** The increased strength and ductility of the present alloy are attributed to the following three dominant mechanisms: 1) dislocation behaviours in the matrix; 2) SF formation in interior PPT (hP24); and 3) TRIP effect in grain-boundary PPT (L1₂). First, Fig. 4b,c shows the deformed substructure of the matrix and that adjacent to the interfaces between the interior PPTs and matrix. High-density dislocations form homogeneous deformation substructures as indicated by a blue arrow, and the high fraction and small interspacing of PPTs lead to massive dislocation interactions in the matrix. Before ageing, the as-quenched SA alloy initially possesses high dislocation density due to inherent characteristics of martensitic transformations. This initial high-density tangled dislocation has limited capability of work hardening as represented in Fig. 3a,b. However, the reduction in dislocation density due the ageing treatment (24H alloy) increases the mean free path of dislocations (MFP). This increased MFP allows uniform dislocation glides at a certain regular spacing (see Fig. 4b,c) and consequent

high ductility. However, it cannot be concluded that this mechanism is solely dominant in strengthening and ductilisation due to massive interior PPTs with the average interparticle spacing of ~58 nm, which limits the substantial increase of MFP. Nevertheless, the uniform dislocation glides and homogeneous deformation substructures contribute to preventing premature cracking in ultrahigh-strength alloys.

Question #1-3> In page 6, the authors defined the driving force for the formation of hP24, i.e., the free energy difference between the parent phase (bcc) and hP24 structure. The composition used for the calculation is the stoichiometric composition of the involved precipitates phase, rather than the alloy composition. As such, the DFT calculation might be able to explain origins of the TRIP effect of the metastable phase, but cannot reveal any direct relationship with the designed alloy compositions. That is, from alloy design point of view, these calculations are insufficient to provide any hints.

Reply #1-3> We appreciate that the comment allows us to explain specific relationships of alloy compositions with precipitate compositions calculated by DFT. Our primary objective was to design a metastable ordered-structure phase as a precipitate. We targeted M_3V -type (A_3B -type with M: Fe, Co, Ni) structures reported from Liu et al. [Ref. 46], as the structures vary from ordered fcc to hcp depending on the electron concentration (e/a, valence electrons per atom) adjusted by M composition. Ni having the highest number of valence electrons was excluded from the candidate as it can result in high hexagonality and the absence of cubic structure. Therefore, we chose Fe and Co as candidates for the M site to target the metastable intermetallic phase between Co_3V and Fe_3V . Stoichiometric compositions from Co_3V to Fe_3V were considered in the DFT calculation, resulting in the lowest energy gap between $L1_2$ and hP24 structure at $Co_2Fe_1V_1$.

However, the calculated composition represents that of precipitate, which cannot reveal any direct relationship with the designed alloy compositions as the reviewer commented. We entirely agree with it. In fact, we had designed the alloy composition via CALPHAD approaches via thermodynamic calculations to form the aimed $\text{Co}_2\text{Fe}_1\text{V}_1$ precipitates. At the first submission, we omitted details of thermodynamic calculations to construct a concise manuscript and also highlight the narrowest energy gap. The calculation details are as follows and have been added to the supplementary materials (Supplementary Fig. 2 and 3).

As aforementioned, three elements of Fe, Co, and V are the candidates for our strategy. The phase diagrams under fixed V content at 10 at% and 20 at% are shown in Supplementary Fig. 2. For the $\text{Fe}_x\text{Co}_{80-x}\text{V}_{20}$ phase diagram, the sigma (σ) phase is present along the Fe-rich regions, which is likely to cause embrittlement. In addition, sufficient Fe is necessary in order to obtain martensitic microstructure. Thus, the V content of 10 at% was considered to avoid brittle σ phase and also to obtain aimed M_3V phase. At 10% V, the next step was determining the proportion of Fe and Co. The calculations in Supplementary Fig. 3 demonstrate that more abundant Co leads to a massive fraction of M_3V . Interestingly, the M_3V phase has the composition aimed $\text{Co}_2\text{Fe}_1\text{V}_1$ in the $\text{Fe}_{50}\text{Co}_{40}\text{V}_{10}$ alloy composition. With increasing the nominal Co content in the alloy, the M_3V phase possesses more Co content (>50 at%), which deviates from the targeted $\text{Co}_2\text{Fe}_1\text{V}_1$. Thus, the $\text{Fe}_{50}\text{Co}_{40}\text{V}_{10}$ was chosen as the alloy composition in order to obtain the aimed $\text{Co}_2\text{Fe}_1\text{V}_1$ precipitate for M_3V precipitate and also to form fully martensitic structures from a single fcc phase at a high-temperature range (>900 °C).

Based on the calculation results, the initial heat treatments were conducted at 1000 °C for 1 h and subsequent ageing at 550 °C for 24 h. We additionally modified the heat-treatment conditions to optimise the mechanical properties; the alloys were annealed at 900 °C for 10 min in order to obtain a smaller prior austenite (fcc) grain size and fully martensitic structure

(bcc) after quenching. Thus, the annealing and ageing temperatures slightly deviate from the equilibrium temperature (920 °C and 546 °C, respectively).

In addition, we conducted additional DFT calculations with the actual composition of the precipitate to ensure the trend of the phase stability between the different crystal structures. The stability of L₁₂ and hP24 was confirmed to intersect between 550–600 K (exactly at 578 K), where L₁₂ structure is more stable at ageing temperature and hP24 at room temperature, respectively. We have included the additional results in Supplementary Fig. 1 and Table 4. Therefore, the following sentences, figures, table, and reference have been added to explain the alloy design approaches.

Results

- **Alloy design.** ...in varying compositions of (Fe,Co)₃V₁ (see details of computational methodology and additional data in Methods and Supplementary Fig. 1). ...
- ...at typical temperatures for the ageing treatment. Then, we performed thermodynamic calculations to form the desired precipitates in the Fe–Co–V ternary system (see details in Methods and Supplementary Fig. 2 and 3). Thus, FeCo_{0.8}V_{0.2} was selected in order to obtain fully martensitic microstructure after quenching as the matrix and form precipitates (PPTs) of the desired M₃V phase (M: Fe, Co) after ageing without other phases causing embrittlement. To fabricate...

Methods

- **Alloy design & fabrications.** As aforementioned, three elements of Fe, Co, and V are the candidates for our strategy. To determine the alloy composition to embody the desired precipitates, we performed thermodynamic calculations based on CALPHAD approaches using Thermo-Calc software with a TCFE2000 database and its upgraded version^{58–61}. The phase diagrams under fixed V content at 10 at% and 20 at% are shown

in Supplementary Fig. 2. For the $\text{Fe}_x\text{Co}_{80-x}\text{V}_{20}$ phase diagram, the sigma (σ) phase is present along the Fe-rich regions, which is likely to cause embrittlement. As mentioned in the manuscript, however, sufficient Fe is necessary in order to obtain martensitic microstructure. Thus, the V content of 10 at% was considered to avoid brittle σ phase, and also to obtain aimed M_3V phase in the martensitic matrix. At 10% V, in determining the proportion of Fe and Co, calculation results in Supplementary Fig. 3 demonstrate that more abundant Co leads to a massive fraction of M_3V . Interestingly, the M_3V phase has the composition aimed $\text{Co}_2\text{Fe}_1\text{V}_1$ in the $\text{Fe}_{50}\text{Co}_{40}\text{V}_{10}$ alloy composition. Further increase of nominal Co content leads to more Co content (>50 at%) in the M_3V phase, which deviates from the targeted $\text{Co}_2\text{Fe}_1\text{V}_1$. Thus, considering the calculation results and the possibility of the formation of brittle sigma phase in the abundant V composition, we selected $\text{Fe}_{50}\text{Co}_{40}\text{V}_{10}$ as a bulk alloy composition with relatively higher Fe and Co content compared to that of V. This composition is expected to obtain the aimed $\text{Co}_2\text{Fe}_1\text{V}_1$ precipitate for M_3V precipitate and also to form fully martensitic structures from single fcc phase at a high-temperature range (>900 °C).

References

- [58.] TCFE2000: The Thermo-Calc Steels Database, upgraded by B.-J. Lee, B. Sundman at KTH, (KTH, Stockholm, 1999).
- [59.] Choi, W. M. *et al.* A Thermodynamic Modelling of the Stability of Sigma Phase in the Cr-Fe-Ni-V High-Entropy Alloy System. *J. Phase Equilibria Diffus.* **39**, 694–701 (2018).
- [60.] Choi, W. M. *et al.* A thermodynamic description of the Co-Cr-Fe-Ni-V system for high-entropy alloy design. *Calphad Comput. Coupling Phase Diagrams Thermochem.* **66**, (2019).

[61.] Do, H. S., Choi, W. M. & Lee, B. J. A thermodynamic description for the Co–Cr–Fe–Mn–Ni system. *J. Mater. Sci.* **57**, 1373–1389 (2022).

Supplementary Fig. 1. Temperature dependence of the Gibbs energy of candidate ordered precipitates (hP24, L12, and D019) with respect to the disordered bcc solid solution approximated based on the Debye–Grüneisen model. The results for different compositions (a Co₃V₁, b Co₂Fe₁V₁, c Co_{1.5}Fe_{1.5}V₁, d Co₁Fe₂V₁, e Fe₃V₁, f, g experimentally obtained composition of the precipitates provided in Supplementary Table 1 and 4) are presented.

Calculated phase diagram of $\text{Fe}_x\text{Co}_{90-x}\text{V}_{10}$ and $\text{Fe}_x\text{Co}_{80-x}\text{V}_{20}$

- **Supplementary Fig. 2. Thermodynamic calculation results (CALPHAD) of Fe-Co-V system with fixed V compositions. Phase diagrams at a $\text{Fe}_x\text{Co}_{90-x}\text{V}_{10}$ and b $\text{Fe}_x\text{Co}_{80-x}\text{V}_{20}$.**

- **Supplementary Fig. 3. Thermodynamic calculation results of phase fraction vs. temperature and detailed M_3V composition. Results for alloy composition of a $\text{Fe}_{50}\text{Co}_{40}\text{V}_{10}$, b $\text{Fe}_{40}\text{Co}_{50}\text{V}_{10}$, and c $\text{Fe}_{30}\text{Co}_{60}\text{V}_{10}$.**

- **Supplementary Table 4. Mole fraction (X_i) and number (N_i) of elements in the supercell used for DFT calculations of alloys with different compositions. Experimentally determined target compositions (Exp.L12 and Exp.hp24) are given in parentheses (Supplementary Table 1).**

Alloy	X_{Co}	X_{Fe}	X_V	N_{Co}	N_{Fe}	N_V
Co ₃ V ₁	0.75	0	0.25	36		12
Exp.L12	0.54167 (0.538)	0.20833 (0.218)	0.25 (0.244)	26	10	12
Exp.hp24	0.52083 (0.517)	0.22917 (0.237)	0.25 (0.246)	25	11	12
Co ₂ Fe ₁ V ₁	0.5	0.25	0.25	24	12	12
Co _{1.5} Fe _{1.5} V ₁	0.375	0.375	0.25	18	18	12
Co ₁ Fe ₂ V ₁	0.25	0.5	0.25	12	24	12
Fe ₃ V ₁	0	0.75	0.25		36	12

Question #1-4> The authors pointed out that boundary precipitates is L12 rather than hp24 because of the difference in both lattice misfit and variant amount. First, the close-packed plane for L12 and hp24 should be identical and they are both the parallel plane in K-S and Burgers OR. Therefore, I guess the misfit of the L12-matrix and hp24-matrix is very similar. Also, the authors should provide the basic data such as lattice parameter, used for the misfit calculation. Second, Figure 2 shows that boundary precipitates mainly grow into one side of grain boundary with obvious orientation relationship. According to the description that they maintain semi-coherent interfaces with both side matrix, the orientation relationship with the other side of grain should be also provided.

Reply #1-4> We sincerely appreciate the helpful comment. Lattice parameters used for lattice misfit calculations were obtained from the XRD analysis; 2.866 Å for the bcc matrix; 3.576 Å

for the L1₂; 4.942 Å and 11.957 Å for the *a* and *c* of hP24, respectively. L1₂ and hP24 exhibit different lattice constants because they have different ordered structures, *i.e.*, ordered cubic and ordered hexagonal structures. Therefore, the existence of a close-packed plane in both structures does not directly relate to the similarity of the lattice constants, unlike disordered structures.

The lattice misfits between the matrix with L1₂ and hP24 are 1.84% and 0.46%, respectively, as provided in the manuscript. The lattice misfits between the precipitates and the matrix were estimated by using the equation $\delta = 2(d_{PPT} - d_{matrix}) / (d_{PPT} + d_{matrix})$, where d_{PPT} and d_{matrix} are the lattice constant of hP24 or L1₂ and bcc martensite. The K-S OR is based on the cubic-cubic relationship, while Burgers OR is based on a hexagonal-cubic relationship. Therefore, d_{hP24} is considered as lattice constant along *a* axis of hP24 and d_{bcc} as $\sqrt{3}$ -fold of the lattice constant of bcc martensite. The following sentences have been revised in the Methods part.

Methods

- **Determination of lattice parameter and misfit.** Lattice parameters used for determining lattice misfit were obtained from the XRD analysis; 2.866 Å for the bcc matrix; 3.576 Å for the L1₂; 4.942 Å and 11.957 Å for the *a* and *c* of hP24, respectively. The lattice misfits between the precipitates and the matrix were estimated by using the equation $\delta = 2(d_{PPT} - d_{matrix}) / (d_{PPT} + d_{matrix})$, where d_{PPT} and d_{matrix} are the lattice constant of hP24 or L1₂ and bcc martensite. The K-S OR is based on the cubic-cubic relationship, while Burgers OR is based on a hexagonal-cubic relationship. Therefore, d_{hP24} is considered as lattice constant along *a* axis of hP24 and d_{bcc} as $\sqrt{3}$ -fold of the lattice constant of bcc martensite.

Regarding the orientation relationship (OR) between L1₂ and matrix at both side grains, our statement was misleading and should have been carefully examined. It has been well-established that a precipitate at grain boundary has a rational OR, *e.g.*, K-S, with an adjacent grain, while the interface is incoherent with the other adjacent grain and is highly dependent on the grain boundary characteristics. However, it was also shown that precipitates at grain boundary become partially coherent by the formation of ledges and misfit compensating defects, as they try to reduce the increment of interfacial energy at most. Therefore, both the intragranular precipitates with a highly faulted structure and the intergranular precipitates showing K-S OR with an adjacent grain are expected to minimise the interfacial energy.

In this regard, we have conducted further TEM analyses to investigate the interface structure between grain-boundary precipitate and the other adjacent grain. As expected, the irrational OR was observed where $BD_{\text{bcc}} = \langle 111 \rangle$ and $BD_{\text{L12}} \sim \langle 114 \rangle$, respectively; however, the HRTEM image (Supplementary Fig. 7) shows an interesting contrast from the phase interface with partially coherent bonding and its arrangement changes with the curved interface. A similar interface structure was observed in three different precipitates. This might be related to the formation of ledges and misfit compensating defects. It is also noted that the activation energy increased with increasing the tilt angle between the low-energy interface boundary and the original matrix grain boundary, when the tilt angle is below the critical value. The closest $\{111\}_{\text{fcc}}$ to the matrix grain boundary ($\sim\{011\}_{\text{bcc}}$) was selected as a low-energy interface. The observed flat interfaces in the present study (Fig. 2j,m) are parallel to the $\{011\}_{\text{bcc}}$, and thus, it seems to be the low-energy interface, while the crystal structure of precipitate and matrix is different from the reference [Ref. 32]. Although further systematic studies are definitely required to fully understand the precipitation behaviour and the interface structure depending on grain boundary characteristics, which is beyond the scope of this study, we have revised the discussion on the precipitation behaviour and added relevant references as follows.

Results

Microstructure and precipitation behaviour. ...experimental findings. Notably, PPTs possess two different crystal structures depending on the nucleation sites. The interior hP24 PPTs develop Burgers OR with the matrix, while the grain-boundary L1₂ PPTs develop K-S OR with the matrix. Based on the ORs of each PPT, the measured lattice misfits exhibited a value of 0.46% for hP24 and 1.84% for L1₂. PPTs having low-energy interfaces, *e.g.*, Burgers and K-S ORs, exhibit flat interfaces leading to rod-shaped morphology for hP24 and polygonal shape for L1₂ and both interfaces are parallel to the $\{011\}_{\text{bcc}}$. It has been well-established that a PPT at grain boundary has a rational OR, *e.g.*, K-S, with an adjacent grain, while the interface is incoherent with the other adjacent grain and highly dependent on the grain boundary characteristics²⁹. However, it was also shown that PPTs at grain boundary become partially coherent by formations of ledges and misfits compensating defects³⁰, as they try to reduce the increment of interfacial energy at most. It was also noted that the activation energy increases with increasing the tilt angle between the low-energy interface and the original matrix grain boundary, when the tilt angle is below the critical value³¹. In other words, the closest $\{111\}_{\text{fcc}}$ to the matrix grain boundary ($\sim\{011\}_{\text{bcc}}$) is selected as a low-energy interface³². Therefore, both the interior PPTs with a highly faulted structure and the grain-boundary PPTs showing K-S OR with an adjacent grain are expected to minimise the interfacial energy. While it is confirmed that L1₂ develops K-S OR with an adjacent grain (Fig. 2o), the interface structure with the other adjacent grain does not show exact K-S OR. As expected, the irrational OR was observed where the beam directions (BDs) were $\text{BD}_{\text{bcc}} = \langle 111 \rangle$ and $\text{BD}_{\text{L12}} \sim \langle 114 \rangle$, respectively; however, the HRTEM image (Supplementary Fig. 7) shows an interesting contrast from the phase interface with partially coherent bonding and its arrangement changes

with the curved interface. This might be related to the formation of ledges and the misfit compensating defects to minimise the interfacial energy³⁰.

To further explain the origin of the precipitation behaviour of the two phases, a selection of structure based on electron concentrations (e/a) of PPTs and their heterogeneous nucleation and growth were further considered. According to the theory based on e/a from Liu et al.³³, the M_3V phase begins to show hexagonality mixed with a cubic crystal structure over e/a of 7.89. The e/a value exhibits ~ 7.81 for our actual composition of the present PPTs, which is lower than the critical concentration forming hexagonality. Therefore, it is likely for the PPTs to form cubic ordered structure, *i.e.*, $L1_2$. However, in the process of heterogeneous nucleation and growth of PPTs, those nucleating on dislocations are dominated by the strain-field effect where the lattice misfit becomes a critical factor³⁴. The interior PPTs would accommodate numerous stacking faults and transform in the direction of the hP24 structure during growth to minimise the misfit ($L1_2$ -bcc: 1.84% vs. hP24-bcc: 0.46%). The narrow energy stability gap between two phases also seems to allow the formation of SFs and local hP24 structure. On the other hand, the PPTs at grain boundaries consume the prior fcc grain boundaries and forms the semi-coherent interface with an adjacent grain to lower the interface energy. This well corresponds to a conventional heterogeneous precipitation mechanism at grain boundaries. When $L1_2$ forms a semi-coherent interface with an adjacent grain, it seems that the formation of SFs does not further reduce the energy, but the locally flat interfaces form with the other adjacent grain, resulting in the polygonal-shaped PPT aforementioned.

- **Supplementary Fig. 7. Phase interfaces between the matrix and grain-boundary L12 precipitates of the near-rational OR side. a** BF image, **b** inverse fast Fourier transform (IFFT) spectra, and **c** enlarged IFFT spectra in **b**.

References

- [29.] Smith, C. S. Microstructure 1952 Campbell, Edward, Demille Memorial lecture. *Trans. ASM* **45**, 533-575 (1953).
- [30.] Furuhashi, T. & Maki, T. Interfacial structure of grain boundary precipitate in a Ni-45 mass%Cr alloy. *Mater. Trans. JIM* **33**, 734-739 (1992).
- [31.] Lee, J. K. & Aaronson, H. I. Influence of faceting upon the equilibrium shape of nuclei at grain boundaries-I. Two-dimensions. *Acta Metall.* **23**, 799-808 (1975).
- [32.] Adachi, Y., Hakata, K. & Tsuzaki, K. Crystallographic analysis of grain boundary Bcc-precipitates in a Ni-Cr alloy by FESEM/EBSD and TEM/Kikuchi line methods. *Mater. Sci. Eng. A* **412**, 252-263 (2005).
- [34.] Furuhashi, T. & Maki, T. Variant selection in heterogeneous nucleation on defects in diffusional phase transformation and precipitation. *Mater. Sci. Eng. A* **312**, 145-154 (2001).

Question #1-5> This paper mainly focused on semi-coherent precipitates. Given the large size, is the interface still semi-coherent?

Reply #1-5> We thank the reviewer for this question, allowing us to elaborate on the semi-coherent interface of the precipitate. As mentioned in **Reply #1-4**, it is hard to determine that all precipitates maintain semi-coherent interfaces under prolonged ageing conditions. Most other maraging alloys are subjected to ageing less than 10 h; thus, precipitates in the present alloy are relatively larger compared to others. In other words, nanoprecipitates in pre-existing maraging alloys exhibit few nanometre sizes, while the interior hP24 precipitates have an average length and diameter of ~160 nm and ~25 nm, respectively. In order to unravel the uncertainty of interfaces, we have performed additional TEM analyses, as shown in HRTEM images in Supplementary Fig. 5, showing two precipitates intersecting one another with interfaces of the matrix/precipitates. The provided image shows an unclear and very diffused interface where it is hard to identify the exact boundary owing to the highly faulted structure of the hP24 precipitate. Although facing a hurdle in direct observation, the FFT images support and provide information that the interface is semi-coherent on $\langle\bar{1}11\rangle_{bcc} // \langle 11\bar{2}0\rangle_{hcp}$ direction indicated by a yellow arrow. The following sentences and HRTEM image have been revised and added to support the semi-coherent interfaces of hP24.

Results

- **Microstructure and precipitation behaviour.** ...because of the high density of SFs (Fig. 2k,l). For large-sized PPTs, high-resolution TEM (HRTEM) and FFT images (Fig. 2l and Supplementary Fig. 5) confirm the...

- **Supplementary Fig. 5. High-resolution transmission electron microscopy (HRTEM) image of two precipitates intersecting within the matrix. Fast Fourier-transform (FFT) images for each spot are presented where the pattern clarity varies with site. Inverse FFT of the red box spot shows faulted structure of hP24 precipitate. The overall pattern indicates Burgers orientation relationship in $\langle \bar{1}11 \rangle_{\text{bcc}} // \langle 11\bar{2}0 \rangle_{\text{hcp}}$ direction.**

As well as the additional TEM works, we have conducted further ageing treatment for 1 week to investigate whether the semi-coherent interfaces are still valid for very larger precipitates (Supplementary Fig. 6). The average length and diameter of ~ 160 nm and ~ 25 nm of 24H alloy increase to ~ 300 nm and ~ 57 nm. Interestingly, although both length and diameter are expected to increase sufficiently, their increments are not considerable, resulting from the low-energy characteristics of semi-coherent interfaces. Besides, the flat facets are still maintained along the longitudinal side of hP24 and also at grain-boundary L1₂. Thus, this result

supports the retention of semi-coherent interfaces even for large precipitates to a certain degree. It is worthwhile that the further aged alloy shows decreased strength but increased ductility, while their amounts are not significant despite the increase of ageing time from 1 day to 7 days. The following sentence has been added to support the semi-coherent interfaces of hP24.

Results

- **Microstructure and precipitation behaviour.** ...they have a diffused phase interface because of the SFs. These semi-coherent interfaces enable the PPT to maintain the nanometre size after further ageing up to 1 week (Supplementary Fig. 6). Unlike the grain interior hP24, the PPTs along...

- **Supplementary Fig. 6. Comparison of microstructures and tensile properties for the alloys aged for 1 day (24H alloy) and 1 week.**

Question #1-6> No scale bar in all IFFT and SAED patterns. Without it, a pattern can be identified to be many phases.

Reply #1-6> We appreciate the helpful comment where we can characterise the phases more precisely. Scale bars have been added in all FFT and SAED patterns in all related figures, including main and supplementary figures.

Question #1-7> Figure 3c, data for recently developed maraging steels should be included. It would be nice to discuss possible difference of underlying mechanisms involved.

Reply #1-7> We appreciate the constructive comments to compare and highlight our approaches with existing ultrastrong alloys. The data for recently developed and also conventional maraging steels have been added in Figure 3c. The region of ultrastrong alloys, *i.e.*, maraging steels group, has been additionally magnified in order to clarify the difference. The tensile properties comparable to our work are reported in maraging steels by Jiang et al. [Ref. 2], which exhibits uniform elongation of 4%. The two alloys (Jiang's and this work) possess similar tensile properties; however, they show very different mechanisms. The primary work hardening mechanism in Jiang's work is slip-band refinements induced by extremely fine and nanosized shearable precipitates having coherent interfaces with minimal lattice misfit. In contrast, the present precipitates have relatively larger size and semi-coherent interfaces, which occurs the Orowan bowing mechanisms with matrix dislocations. Besides, deformable hP24 and transformable L1₂ precipitates play an essential role in both strengthening and ductilisation. Therefore, the following sentences and figure have been revised and added relevant references as follows.

Results

- **Mechanical properties.** ...reported for ultrastrong precipitation-strengthened HEAs, MEAs, and maraging steels. The inset in Fig. 3c is the magnified region of ultrahigh strength maraging group to clearly distinguish the properties. When it comes to ultrahigh strength metallic materials with strength reaching near 2 GPa, most of them show uniform ductility of less than 2%, while the present 24H alloy reaches 4%.
- **Discussions.** In summary, we demonstrate a design strategy resulting in an ultrahigh strength of ~2 GPa and acceptable uniform elongation of ~4.0% through deformable hP24 and transformable L1₂ PPTs. ...

- **Figure 3. Room-temperature mechanical properties of the alloys.** ... c Comparison of yield strength versus uniform elongation for the FeCo_{0.8}V_{0.2} MEAs and other single or multiphase high-/medium-entropy alloys and maraging steels.

Supplementary References

- [27.] Schnitzer, R. *et al.* Influence of reverted austenite on static and dynamic mechanical properties of a PH 13-8 Mo maraging steel. *Mater. Sci. Eng. A* **527**, 2065–2070 (2010).
- [28.] Jiang, S. H. *et al.* Strain hardening mediated by coherent nanoprecipitates in ultrahigh-strength steels. *Acta Mater.* **213**, (2021).
- [29.] Sato, K. Improving the toughness of ultrahigh strength steel PhD thesis, Univ. California, Berkeley (2002).
- [30.] Kim, Y. K., Kim, K. S., Song, Y. B., Park, J. H. & Lee, K. A. 2.47 GPa grade ultra-strong 15Co-12Ni secondary hardening steel with superior ductility and fracture toughness. *J. Mater. Sci. Technol.* **66**, 36–45 (2021).
- [31.] Niu, M. *et al.* Precipitate evolution and strengthening behavior during aging process in a 2.5 GPa grade maraging steel. *Acta Mater.* **179**, 296–307 (2019).

Reviewer #2 (Remarks to the Author):

This article presents work assessing the maraging effect in a FeCo_{0.8}V_{0.2} alloy. It comprises some interesting analysis and is a good demonstration of a joint modelling & experiment approach. Unfortunately, I'm not convinced of the significance of the phase transformation mechanisms proposed, nor that the alloy developed has particularly special mechanical properties. I have the following specific comments (in order of appearance in the text):

Question #2-1> Introduction text: “preliminary” should be primarily?

Reply #2-1> We appreciate the reviewer for revising the expression more appropriately. The text has been revised as follows:

- **Introduction.** ... The technological importance of martensite **primarily** comes from its high strength based on hierarchy substructures...

Question #2-2> To say that the technological importance of martensite originates from maraging is an odd statement. Most martensitic microstructures in use today (i.e., those in steels) are not age hardened, but rather just tempered (softened). It is tempering that allows martensite to be useful in most cases. The statement seems to contradict the statement later in the paragraph, which says that current maraging alloys aren't applied much (indicating that maraging isn't very technologically important).

Reply #2-2> We agree with the reviewer's comment, and thank the reviewer very much for pointing out the odd statement. The maraging alloys are still technologically important, while those do not result from the inherent characteristics of martensite, as the reviewer commented, but they are associated with a special class of very low-carbon steels. Maraging alloys are not hardened by carbon or carbides but by precipitations of intermetallic compounds, which allows

for achieving combinations of high strength and toughness while maintaining relatively high ductility. Therefore, we have revised the following sentences and added reference.

- **Introduction.** ...based on hierarchy substructures, while in most carbon steels the martensitic microstructures are subjected to tempering that assigns greater toughness by increasing ductility but decreasing strength. As a special class of very low-carbon steels hindering formations of brittle carbides, maraging alloys (martensite+ageing) achieve an outstanding combination of strength and toughness while maintaining relatively high ductility. The maraging can restore the ductility with the reduction of lattice defects formed during martensitic transformation and exploit an additional hardening effect through the formation of nanosized intermetallic precipitates²⁻⁷, instead of various carbides as in carbon-bearing tempered martensite. However, the introduced large coherency strains with heterogeneous distribution of semi-coherent precipitates may lead to crack initiation as a double-edged sword⁸⁻¹⁰. Besides, the uniform ductility is limited to approximately 2% due to the limited work hardening in commercial maraging alloys^{6,7,11} exhibiting a yield strength of 2 gigapascals (GPa) or higher, thus requiring further enhancement of both strength and work hardening for their widespread applications.

References

- [11.] Kim, Y. K., Kim, K. S., Song, Y. B., Park, J. H. & Lee, K. A. 2.47 GPa grade ultra-strong 15Co-12Ni secondary hardening steel with superior ductility and fracture toughness. *J. Mater. Sci. Technol.* **66**, 36–45 (2021).

Question #2-3> More explanation is needed when the calculations on 50Co-25Fe-25V are introduced. The reader has no idea what the expected microstructure of FeCo_{0.8}V_{0.2} should be, and therefore why calculations of Co₂FeV are important.

Reply #2-3> We sincerely thank the reviewer for the critical question. We had focused only on the precipitates but given relatively little attention to designs of alloy composition and overall microstructures. In this study, we started with a hard martensite matrix as a base microstructure to obtain high yield strength, and further introduced semi-coherent intermetallic phases, which underwent dynamic phase transformation. As described in **Reply #1-3**, firstly, our primary objective was to design a metastable ordered-structure phase as a precipitate. We targeted M₃V-type (A₃B-type with M: Fe, Co, Ni) structures from Liu et al. [Ref. 33], as the structures vary from ordered fcc to hcp depending on the electron concentration (e/a, valence electrons per atom) adjusted by M composition. Ni having the highest number of valence electrons was excluded from the candidate as it can result in high hexagonality and the absence of cubic structure. Therefore, we chose Fe and Co as candidates for the M site to target the metastable intermetallic phase between Co₃V and Fe₃V. Stoichiometric compositions from Co₃V to Fe₃V were considered in the DFT calculation, consequently resulting in the lowest energy gap between L1₂ and hP24 structure at Co₂Fe₁V₁. However, the calculated composition represents that of precipitate, which cannot reveal any direct relationship with the designed alloy compositions and overall microstructures as the reviewer commented. In order to form precipitates having Co₂Fe₁V in the martensite matrix, the following CALPHAD approaches had been performed via thermodynamic calculations.

Please kindly refer to Supplementary Fig. 2 and 3, and **Reply #1-3** for details of CALPHAD approaches. At the first submission, we omitted details of thermodynamic calculations to construct a concise manuscript and also highlight the narrowest energy gap. Based on the phase

diagram under fixed V content at 10 at% and 20 at%, $\text{Fe}_x\text{Co}_{90-x}\text{V}_{10}$ was selected to obtain aimed M_3V phase and also to avoid sigma (σ) phase which is likely to cause embrittlement. In $\text{Fe}_x\text{Co}_{90-x}\text{V}_{10}$ compositions (Supplementary Fig. 2), sufficient Fe is necessary in order to obtain martensitic microstructure, while more abundant Co leads to a massive fraction of M_3V . Interestingly, the M_3V phase has the composition aimed $\text{Co}_2\text{Fe}_1\text{V}_1$ in the $\text{Fe}_{50}\text{Co}_{40}\text{V}_{10}$ alloy composition. The M_3V phase possesses more Co content (>50 at%) with increasing the nominal Co content, which deviates from the targeted $\text{Co}_2\text{Fe}_1\text{V}_1$. Thus, the $\text{Fe}_{50}\text{Co}_{40}\text{V}_{10}$ was chosen as the alloy composition in order to obtain the aimed $\text{Co}_2\text{Fe}_1\text{V}_1$ precipitate for M_3V precipitate and also to form fully martensitic structures from single fcc phase at a high-temperature range (>900 °C). Therefore, the following sentences to explain the design of alloy composition, precipitate composition, and overall microstructures have been revised as follows.

Results

- **Alloy design.** ...in varying compositions of $(\text{Fe,Co})_3\text{V}_1$ (see details of computational methodology and additional data in Methods and Supplementary Fig. 1).
- ...at typical temperatures for the ageing treatment. Then, we performed thermodynamic calculations to form the desired precipitates in the Fe–Co–V ternary system (see details in Methods and Supplementary Fig. 2 and 3). Thus, $\text{FeCo}_{0.8}\text{V}_{0.2}$ was selected in order to obtain fully martensitic microstructure after quenching as the matrix and form precipitates (PPTs) of the desired M_3V phase (M: Fe, Co) after ageing without other phases causing embrittlement. To fabricate...

Methods

- **Alloy design & fabrications.** As aforementioned, three elements of Fe, Co, and V are the candidates for our strategy. To determine the alloy composition to embody the desired precipitates, we performed thermodynamic calculations based on CALPHAD

approaches using Thermo-Calc software with a TCFE2000 database and its upgraded version⁵⁸⁻⁶¹. The phase diagrams under fixed V content at 10 at% and 20 at% are shown in Supplementary Fig. 2. For the $\text{Fe}_x\text{Co}_{80-x}\text{V}_{20}$ phase diagram, the sigma (σ) phase is present along the Fe-rich regions, which is likely to cause embrittlement. As mentioned in the manuscript, however, sufficient Fe is necessary in order to obtain martensitic microstructure. Thus, the V content of 10 at% was considered to avoid brittle σ phase, and also to obtain aimed M_3V phase in the martensitic matrix. At 10% V, in determining the proportion of Fe and Co, calculation results in Supplementary Fig. 3 demonstrate that more abundant Co leads to a massive fraction of M_3V . Interestingly, the M_3V phase has the composition aimed $\text{Co}_2\text{Fe}_1\text{V}_1$ in the $\text{Fe}_{50}\text{Co}_{40}\text{V}_{10}$ alloy composition. Further increase of nominal Co content leads to more Co content (>50 at%) in the M_3V phase, which deviates from the targeted $\text{Co}_2\text{Fe}_1\text{V}_1$. Thus, considering the calculation results and the possibility of the formation of brittle sigma phase in the abundant V composition, we selected $\text{Fe}_{50}\text{Co}_{40}\text{V}_{10}$ as a bulk alloy composition with relatively higher Fe and Co content compared to that of V. This composition is expected to obtain the aimed $\text{Co}_2\text{Fe}_1\text{V}_1$ precipitate for M_3V precipitate and also to form fully martensitic structures from single fcc phase at a high-temperature range (>900 °C).

References

- [58.] TCFE2000: The Thermo-Calc Steels Database, upgraded by B.-J. Lee, B. Sundman at KTH, (KTH, Stockholm, 1999).
- [59.] Choi, W. M. *et al.* A Thermodynamic Modelling of the Stability of Sigma Phase in the Cr-Fe-Ni-V High-Entropy Alloy System. *J. Phase Equilibria Diffus.* **39**, 694–701 (2018).

- [60.] Choi, W. M. *et al.* A thermodynamic description of the Co-Cr-Fe-Ni-V system for high-entropy alloy design. *Calphad Comput. Coupling Phase Diagrams Thermochem.* **66**, (2019).
- [61.] Do, H. S., Choi, W. M. & Lee, B. J. A thermodynamic description for the Co-Cr-Fe-Mn-Ni system. *J. Mater. Sci.* **57**, 1373–1389 (2022).

Question #2-4> There's no such thing as an fcc L12 phase. L12 is primitive.

Reply #2-4> We appreciate the helpful comment. We have revised the text as follows:

- **Introduction.** ...50Co–25Fe–25V precipitates show an indistinct difference in the phase stability between hP24 (Al₃Pu-type) (ordered hexagonal close-packed (hcp) structure) and L1₂ (ordered fcc structure). ...

Question #2-5> A uniform ductility of 4% doesn't seem very impressive. Should it be seen as so?

Reply #2-5> We appreciate that it allows us to emphasise the novelty of this study. The high strength is obtainable because of the numerous lattice defects which obstruct plastic deformation efficiently withstanding a high-stress level; however, it leads to a drastic drop in uniform ductility concurrently. Thus, the primary purpose for developing such ultrahigh strength materials is to withstand high loads and also to avoid failure by obtaining sufficient uniform ductility. Nevertheless, a limited uniform ductility might lead to catastrophic failure owing to localised deformation in harsh load-bearing applications. The value 4% of uniform ductility may not seem very impressive compared to conventional high-strength alloys. However, when it comes to ultrahigh strength metallic materials with strength reaching near 2

GPa, most of them show uniform ductility of less than 2%. As illustrated in Fig. 3c, the uniform ductility steeply decreases with increasing strength, which is very limited from the point where the strength level exceeds 1.5 GPa. Therefore, the uniform ductility of 4% in the current FeCo_{0.8}V_{0.2} alloy is an impressive result compared to other maraging alloys with a yield strength of approximately 2 GPa.

Although gaining uniform ductility of 4% at near 2 GPa yield strength is impressive indeed, in fact, this mechanical property is not the first to be reported. Most recently developed maraging steel by Jiang et al. [Ref. 2] shown in Fig. 3c has been spotlighted for its superior properties, which exhibit similar properties to this study. The proposed steel exploited dense and homogeneous distributions of nanosized coherent and shearable precipitates, which induces dislocations' planar-slip behaviour and slip-band refinement effects. On the other hand, the present work utilises semi-coherent, non-shearable, and metastable precipitates. The interior hP24 precipitates form Orowan loops in the matrix through the dislocations-precipitate interaction, which at the same time contributes to plasticity with partial dislocation gliding within them. Additionally, the L1₂ precipitates along the grain boundaries go through dynamic phase transformation, providing high work hardening and uniform ductility. Consequently, the complex and interdependent mechanisms result in an impressive combination of strength (~2 GPa) and uniform ductility (~4%). This kind of unique behaviour has never been reported before and therefore gives a novelty to this study. Therefore, the following sentences have been revised as follows.

Results

- **Mechanical properties.** ...reported for ultrastrong precipitation-strengthened HEAs, MEAs, and maraging steels. The inset in Fig. 3c is the magnified region of ultrahigh strength maraging group to clearly distinguish the properties. When it comes to

ultrahigh strength metallic materials with strength reaching near 2 GPa, most of them show uniform ductility of less than 2%, while the present 24H alloy reaches 4%.

Discussion

- In summary, we demonstrate a design strategy resulting in an ultrahigh strength of ~2 GPa and acceptable uniform elongation of ~4.0% through deformable hP24 and transformable L1₂ PPTs. ...

Question #2-6> The technological importance of FeCo_{0.8}V_{0.2} is never discussed. Why chose this alloy? Why these elements?

Reply #2-6> We appreciate the valuable comment on mentioning the technological importance of FeCo_{0.8}V_{0.2} alloy. Generally, maraging alloys are used in load-bearing applications, especially aerospace products such as aircraft landing gear, missile casings, high-performance shafts in jet engines, coil springs, and bolts. Those applications require primarily high yield strength and ductility due to the margin of safety in service as described in **Reply #2-5**. Accordingly, FeCo_{0.8}V_{0.2} alloy was suggested as a model alloy system to implement the requirements with unprecedented metallurgical mechanisms. Details of element selections to obtain martensitic matrix and metastable precipitates are described in **Reply #2-3**.

As well as mechanical properties and associated strengthening-ductilisation mechanisms, the absence of carbon in the present alloy and consequent relatively ductile martensite before ageing exhibit good formability, enabling the homogenised cast ingot to gain a cold-rolling reduction of approximately 80%. In addition, no retained austenite is attainable when quenching to room temperature, which might be attributed to high M_s temperature due to high Co content. In commercial maraging alloys using TRIP effect, such as Aermet 100 and Aermet 340 etc., several hours of deep cryogenic treatment (called DCT) is required before ageing as

the excessive retained austenite significantly decreases the yield strength. In this regard, the current alloy only needs simple heat treatment to obtain superior properties, *i.e.*, solid-solution treatment and post ageing treatment, without any cryogenic treatment or deformation prior to ageing. Therefore, the following sentences have been added to the discussion part.

- **Discussion.** ...TRIP effect in grain-boundary PPTs. These complex metallurgical mechanisms with simple heat treatment are suggested to implement high-performance and load-bearing application requirements. We expect these...

Question #2-7> The alloy design section only very briefly addressed the possibility of transformation between the ordered phases – e.g., it didn't state which transformation was going to be targeted. Also, it's not clear why such a transformation should be beneficial. In TRIP steels, there is a dramatic increase in ductility owing to the considerable shape and volume change associated with the formation of martensite (so you get a lot of hardening and a lot of strain). What about in this case?

Reply #2-7> We appreciate the comments that allow us to provide more evident explanations for the transformation of ordered phases. As the reviewer accurately pointed out, in alloy designs through DFT or CALPHAD approaches, it always remains an unpredictable challenge to implement the calculated properties in real microstructures. In other words, strictly speaking, experimental investigations are essential as it is difficult to predict all microstructures and properties completely. Nevertheless, the cornerstone of our design lies in the diversity of crystal structures of M_3V and their mutual dependence based on stacking sequences originating from L_{12} . The previous report introduced the control of types in M_3V (L_{12} -hP24-D0₂₂) ordered structures through adjusting compositions and electron concentrations [Ref. 33]. In addition, changes in stacking sequences through shearing of L_{12} (in gamma-prime $Ni_3(Al,Ti)$

precipitation hardened nickel-base alloys) resulted in different structures of SnNi₃-type D0₁₉, TiNi₃-type D0₂₄, and VCo₃-type hP24 [Ref. 47]. In the L₁₂ structure, stacking faults are of three basic types: superlattice intrinsic (or extrinsic) faults, antiphase boundary faults, and complex faults. Superlattice intrinsic (S-ISF) and extrinsic (S-ESF) stacking faults are related to L₁₂ through shear displacements of the type $\{111\}1/3\langle 112\rangle$ or climb displacements of the type $\{111\}1/3\langle 111\rangle$, and have characteristics: S-ISF = 4 layers of D0₁₉; S-ESF = 7 layers of D0₂₄; and S-ISF + S-ESF = 10 layers of hP24.

Therefore, combining two ideas of (i) the initial crystal structure can be controlled through compositional configurations and (ii) the stacking faults induced by deformation are able to transform L₁₂ into hexagonal ordered structures is the original concept. In other words, the transformation from L₁₂ to hP24 structures was targeted by comparing the phase stability depending on the composition; Co₃V, Co₂Fe₁V₁, Co_{1.5}Fe_{1.5}V₁, Co₁Fe₂V₁, and Fe₃V. The L₁₂ phase possesses a crystal structure of ordered fcc with a stacking sequence of ‘*abcabcabc...*’ and the hP24 has an ordered-hcp structure with a stacking sequence of ‘*abcacbababc...*’ which exhibits a twin-like formation. Thus, shear displacements and stacking faults by deformation can induce dynamic precipitate transformation as the stacking sequence of the two phases are akin to one another, as illustrated in Supplementary Fig. 9.

Regarding the second comment, the dynamic precipitate transformation in the present study has a certain novelty because the strengthening through deformable hard precipitate has not been reported before. Similar to the concern in **Question #2-12** regarding the work hardening of TRIP effects, the existence of a soft austenite phase in TRIP-assisted steels generally allows a dramatic increase in work-hardening rate and ductility. Although a high fraction of metastable austenite contributes to considerable plasticity, it also accompanies a decrease in yield strength, and thus this kind of microstructure cannot implement a class of ultrahigh strength steel. In

order to maintain the ultrahigh strength level and gain additional ductility, a fraction of the soft phase should keep a minimum, or its morphology should be tuned to possess high strength level and high mechanical stability. As for the represented case, Aermet100 contains 1–6% metastable austenite, which presents as thin foils with no downside to the strength. Although it is hard to exhibit a significant increment of work-hardening rate or ductility in a large scope such as general high-strength TRIP steels, it can contribute to preventing premature failure and improving toughness effectively. Therefore, it is worth mentioning that the TRIP effect in ultrahigh strength alloys exhibits different performances from the conventional high-strength TRIP steels.

In this respect, the present work exploits inherently hard intermetallic phases, which also have no downside to the strength, and their deformable and transformable characteristics provide a two-fold enhancement in strength and ductility via ageing. With the newly demonstrated mechanism, the authors envisage the dynamic precipitate transformation concept to be more developed for designing future structural materials. Please kindly refer to **Reply #2-12** regarding the detailed plastic accommodation mechanisms of the present alloy. Therefore, the following sentences and relevant references have been added as follows.

- **Discussion.** The current dynamic precipitate transformation at boundaries has similar effects to TRIP effects in Mn steels or quenching and partitioning (Q&P) steels that consist of metastable austenite at the boundaries^{41,42}. It is well known that the dynamic phase transformation of the metastable phase in TRIP steels and alloys postpones plastic instability and thus enhances ductility and work hardening in a large scope^{43,44}. The high fraction of metastable austenite contributes to considerable plasticity; however, it also accompanies a decrease in yield strength, and thus this kind of microstructure cannot implement a class of ultrahigh strength steel. In order to

maintain the ultrahigh strength level and gain additional ductility, a fraction of the soft phase should keep a minimum, or its morphology should be tuned to possess high strength level and high mechanical stability. As for the represented case, Aermet100 contains 1–6% metastable austenite, which presents as thin foils with no downside to the strength. Although it is hard to exhibit a significant increment of work-hardening rate or ductility in a large scope such as general high-strength TRIP steels, it can contribute to preventing premature failure and improving toughness effectively, as observed in several studies of Aermet100, PH13-8 Mo, and Mn steels^{40,45,46}. Therefore, it is worth mentioning that the TRIP effect in ultrahigh strength alloys exhibits different performances from the conventional high-strength TRIP steels. In this respect, the present work exploits inherently hard intermetallic phases, which also have no downside to the strength, and their deformable and transformable characteristics provide a two-fold enhancement in strength and ductility via ageing.

The underlying mechanism of dynamic structural changes from $L1_2$ to hP24 can be understood based on the stacking faults pair. As observed in $Ni_3(Al,Ti)$ precipitation-hardened nickel-based alloys, $L1_2$ shearing by matrix dislocations can result in different structures of Sn Ni_3 -type $D0_{19}$, Ti Ni_3 -type $D0_{24}$, and VC O_3 -type hP24⁴⁷. The stacking sequence of $L1_2$ is *ABCABCA...*, whereas that of hP24 is *ABCACBA...*, which exhibits twin-like formation. The stacking sequence of $L1_2$ can be changed to hP24 through the shear displacements of the type $\{111\}1/3\langle 112\rangle$ with superlattice intrinsic (S-ISF) and extrinsic (S-ESF) stacking faults pair⁴⁷. ...

References

- [40.] Schnitzer, R. *et al.* Influence of reverted austenite on static and dynamic mechanical properties of a PH 13-8 Mo maraging steel. *Mater. Sci. Eng. A* **527**, 2065–2070 (2010).
- [45.] Sato, K. Improving the toughness of ultrahigh strength steel PhD thesis (Univ. California, Berkeley, 2002).
- [46.] Luo, H. *et al.* Experimental and numerical analysis on formation of stable austenite during the intercritical annealing of 5Mn steel. *Acta Mater.* **59**, 4002–4014 (2011).

Question #2-8> In Fig. 2a, there are features that do not look like clean bcc grains. Is this the “substructure” that is mentioned? What is meant by “substructure”?

Reply #2-8> We appreciate the comment, allowing us to explain the martensitic microstructure in more detail. As the reviewer pointed out, those are not boundaries of clean bcc grains but are substructures in bcc martensite. The diffusionless martensitic transformation involves crystallographic change with simultaneous and cooperative movement of atoms, *i.e.*, military transformation. The crystal structure transforms from fcc to bcc by shear that produce plastic strain. The resulting strain is not only carried out by macroscopic shape change but also through microscopic scale that does not change the crystal structure. The fine-scale change, accompanied by strain, results in inhomogeneous internal substructures, usually through slip or twinning. Moreover, these substructures are classified as packet, block, and lath depending on their crystallographic features. Likewise, the substructures observed in the SEM images in Fig. 2b are the results of athermal martensitic transformation which spontaneously occurred during the quenching process. The substructure division also depends on the prior austenite grain size. The larger prior austenite grain size accommodates more strain change than the

small grain size and therefore requires more division of substructures. The example of this case is observable in Supplementary Fig. 4. Therefore, the following sentence has been revised to indicate the substructures.

Results

- **Microstructure and precipitation behaviour.** ...consequent coarse-grained prior fcc phase ensure the characteristic martensitic structure, including packet, block, and lath substructures (Supplementary Fig. 4). ...

Question #2-9> It is stated that “the proximity histogram across the PPT and bcc matrix (Fig. 2h) implies that precipitation involving atomic diffusion was not yet completed.” What aspect implies this is the case? The smooth transition in composition? And what is the reasoning?

Reply #2-9> We appreciate the comment regarding the ambiguous expression. We had stated that the atomic diffusion would not complete forming the aimed precipitates yet based on comparing precipitate compositions between the experimental and calculated results. For the reviewer’s concern, we had intended the composition of precipitate, not the smooth transition of composition at the interface. In order to improve readability, the relevant sentence has been removed, and the following sentences have been revised in results part.

Results

- **Microstructure and precipitation behaviour.** ... An APT reconstruction of the 1H alloy clearly reveals the very small size (width of ~4 nm) of the PPTs (Fig. 2h), whereas those in 24H alloy are determined to exhibit average length and diameter of ~160 nm and ~25 nm, respectively (Fig. 2p). The local lattice...

Question #2-10> It is suggested that the tendency of one precipitate to nucleate at grain boundaries compared to the other can be related to the number of variants in the OR. This is an unusual suggestion – can the authors provide evidence of where this has been demonstrated before? It still seems unlikely that good correspondence would be found on both sides of the boundary very often, even when there are lots of possible variants.

Reply #2-10> We are sincerely thankful for the question as it allows us to explain the mechanisms of multiphase precipitation more elaborately. Our suggestion was based on the idea that when a secondary phase nucleates at the boundaries, the interfacial energy has a critical impact on the driving force for nucleation along with the chemical driving force. Precipitates tend to nucleate with the lowest interfacial energy with the surrounding matrix as possible constructing orientation relationship. As in the case of intergranular or interphase precipitates, they possess partially coherent interfaces as much as possible with adjacent matrix grains with ledges and misfit-compensating defects [Ref. A, B]. Variant selections are made to minimise the interfacial energy depending on the orientation relationship between the precipitate and the matrix. Therefore, the orientation relationship type and variant selection are important in determining precipitation behaviour and have been discussed in several reports. For example, in duplex stainless steel, precipitation sites differ where carbide ($M_{23}C_6$) was not observed at the δ/δ ferrite interface while nitride (Cr_2N) was observed owing to orientation relationship types [Ref. C].

Thanks to the reviewer's critical comment, we could discuss and investigate the origin of the precipitation behaviour of the two phases more in detail. According to the electron concentration from Liu et al. [Ref. 33], the M_3V phase begins to show hexagonality mixed with a cubic crystal structure over 7.89. The electron concentration value exhibits ~ 7.81 , when applying our actual composition of the precipitates, which is lower than the critical

concentration forming hexagonality. Therefore, it is likely for the precipitates to form cubic ordered structure, *i.e.*, L1₂. However, in the process of heterogeneous nucleation and growth of precipitates, those nucleating on dislocations are dominated by the strain-field effect where the lattice misfit becomes a critical factor [Ref. 34]. The interior precipitate would accommodate numerous stacking faults and transform in the direction of the hP24 structure in the growth process in order to minimise the misfit (L1₂-bcc: 1.84%, hP24-bcc: 0.46%). The narrow energy stability gap between the two-phase also seems to allow the formation of SFs and local hP24 structure. On the other hand, the precipitate at the grain boundaries consumes the prior austenite grain boundaries and forms the semi-coherent interface with an adjacent grain to lower the interface energy. This is a conventional heterogeneous precipitation mechanism at grain boundaries. When L1₂ forms a semi-coherent interface with an adjacent grain, it seems that the formation of SFs does not further reduce the energy, but the locally flat interfaces with the other adjacent grain were observed, resulting in the polygonal-shaped precipitate (Fig. 2g). Although the further systematic investigation is required for the fundamental understanding of precipitation behaviour, which is beyond the scope of this study, we have revised our manuscript as follows.

Results

Microstructure and precipitation behaviour. ...experimental findings. Notably, PPTs possess two different crystal structures depending on the nucleation sites. The interior hP24 PPTs develop Burgers OR with the matrix, while the grain-boundary L1₂ PPTs develop K-S OR with the matrix. Based on the ORs of each PPT, the measured lattice misfits exhibited a value of 0.46% for hP24 and 1.84% for L1₂. PPTs having low-energy interfaces, *e.g.*, Burgers and K-S ORs, exhibit flat interfaces leading to rod-shaped morphology for hP24 and polygonal shape for L1₂ and both interfaces are

parallel to the $\{011\}_{\text{bcc}}$. It has been well-established that a PPT at grain boundary has a rational OR, *e.g.*, K-S, with an adjacent grain, while the interface is incoherent with the other adjacent grain and highly dependent on the grain boundary characteristics²⁹. However, it was also shown that PPTs at grain boundary become partially coherent by formations of ledges and misfit compensating defects³⁰, as they try to reduce the increment of interfacial energy at most. It was also noted that the activation energy increases with increasing the tilt angle between the low-energy interface and the original matrix grain boundary, when the tilt angle is below the critical value³¹. In other words, the closest $\{111\}_{\text{fcc}}$ to the matrix grain boundary ($\sim\{011\}_{\text{bcc}}$) is selected as a low-energy interface³². Therefore, both the interior PPTs with a highly faulted structure and the grain-boundary PPTs showing K-S OR with an adjacent grain are expected to minimise the interfacial energy. While it is confirmed that L_{12} develops K-S OR with an adjacent grain (Fig. 2o), the interface structure with the other adjacent grain does not show exact K-S OR. As expected, the irrational OR was observed where the beam directions (BDs) were $\text{BD}_{\text{bcc}} = \langle 111 \rangle$ and $\text{BD}_{L_{12}} \sim \langle 114 \rangle$, respectively; however, the HRTEM image (Supplementary Fig. 7) shows an interesting contrast from the phase interface with partially coherent bonding and its arrangement changes with the curved interface. This might be related to the formation of ledges and the misfit compensating defects to minimise the interfacial energy³⁰.

To further explain the origin of the precipitation behaviour of the two phases, a selection of structure based on electron concentrations (e/a) of PPTs and their heterogeneous nucleation and growth were further considered. According to the theory based on e/a from Liu et al.³³, the M_3V phase begins to show hexagonality mixed with a cubic crystal structure over e/a of 7.89. The e/a value exhibits ~ 7.81 for our actual composition of the present PPTs, which is lower than the critical concentration forming

hexagonality. Therefore, it is likely for the PPTs to form cubic ordered structure, *i.e.*, L₁₂. However, in the process of heterogeneous nucleation and growth of PPTs, those nucleating on dislocations are dominated by the strain-field effect where the lattice misfit becomes a critical factor³⁴. The interior PPTs would accommodate numerous stacking faults and transform in the direction of the hP24 structure during growth to minimise the misfit (L₁₂-bcc: 1.84% vs. hP24-bcc: 0.46%). The narrow energy stability gap between two phases also seems to allow the formation of SFs and local hP24 structure. On the other hand, the PPTs at grain boundaries consume the prior fcc grain boundaries and forms the semi-coherent interface with an adjacent grain to lower the interface energy. This well corresponds to a conventional heterogeneous precipitation mechanism at grain boundaries. When L₁₂ forms a semi-coherent interface with an adjacent grain, it seems that the formation of SFs does not further reduce the energy, but the locally flat interfaces form with the other adjacent grain, resulting in the polygonal-shaped PPT aforementioned.

References

- [A.] Zheng, Y., Williams, R. E. A., Viswanathan, G. B., Clark, W. A. T. & Fraser, H. L. Determination of the structure of α - β interfaces in metastable β -Ti alloys. *Acta Mater.* **150**, 25–39 (2018).
- [B.] Hall, M. G., Aaronson, H. I. & Kinsma, K. R. The structure of nearly coherent fcc: bcc boundaries in a CuCr alloy. *Surf. Sci.* **31**, 257–274 (1972).
- [C.] Maetz, J. Y., Douillard, T., Cazottes, S., Verdu, C. & Kléber, X. M23C6 carbides and Cr₂N nitrides in aged duplex stainless steel: A SEM, TEM and FIB tomography investigation. *Micron* **84**, 43–53 (2016).

Question #2-11> I feel Fig. 3 should include some conventional maraging alloys, since many at least meet the properties found in this work (e.g., Aermet 340).

Reply #2-11> We appreciate the helpful suggestion. We have included some conventional and recently developed maraging alloys as revised in Fig. 3c. Please kindly refer to **Reply #1-7** for the details of explanations.

- **Figure 3. Room-temperature mechanical properties of the alloys. ... c** Comparison of yield strength versus uniform elongation for the FeCo_{0.8}V_{0.2} MEAs and other single or multiphase high-/medium-entropy alloys and maraging steels.

Question #2-12> More discussion is needed on the mechanism proposed for the work hardening – presumably, the precipitate phase transformation leads to a harder precipitate than originally? How does this mean that more plastic deformation can be accommodated after the transformation (which is the next sentence)?

Reply #2-12> We appreciate the reviewer for this constructive comment, allowing us to discuss the work hardening mechanism in more detail. The conventional TRIP steels that accommodate phase transformation from fcc to bcc (or bct, α' martensite) exhibit the following mechanisms contributing to the increased work hardening. The deformation accumulates strain energy in the soft fcc phase, which is later consumed for phase transformation. Then, the diffusionless transformation of fcc to bcc eventually leads to a harder phase, and the local hard phase prevents initiation of plastic instability. Furthermore, the volume expansion during the transformation creates geometrically necessary dislocations (GND) along the interfaces, which grants additional work hardening. Another example of TRIP effects is associated with fcc to hcp martensite transformation. Similar to twinning-induced plasticity (TWIP), partial dislocation glides on slip planes are known to be required for deformation-induced fcc to hcp transformation. Therefore, the fcc to hcp martensite transformation introduces phase boundaries including stacking faults, and thus, effectively reduces the mean free path of dislocation and even hinders the activation of secondary slip system due to the limited number of slip systems in the hcp structure. This mechanism, therefore, significantly contributes to work hardening through the dynamic Hall–Petch effect. For the case of our alloy, the characteristics of transformation consuming the accumulated strain energy and introducing phase interfaces are similar to the TRIP associated with fcc to hcp transformation. Furthermore, the cubic to hexagonal transformation in the A_3B -type structure significantly increases frictional stress due to the interplanar-locking effects. The precipitates themselves deform by adopting partial dislocations and stacking faults, corresponding to the accommodating mechanism of plastic deformations. Therefore, unlike the conventional ordered precipitates or carbides that contribute to only strengthening, precipitates in the present alloy are able to implement both strengthening and ductilisation mechanisms. Thus, the following sentences have been revised in the discussion part and relevant references have been added.

- **Discussion.** In this respect, the glide of partial dislocations enables L1₂ to hP24 precipitate transformation during deformation, introducing additional phase boundaries and SFs. The introduced interfaces effectively reduce the mean free path of dislocations and even hinder the activation of secondary slip systems due to the limited number of slip systems in the hcp structure^{13,48}. This mechanism, therefore, significantly contributes to work hardening through the dynamic Hall–Petch effect. The phase transformation also relieves the strain energy accumulated during tensile deformation, enabling further plastic deformation to be accommodated. Furthermore, the cubic to hexagonal transformation in the A₃B-type structure is known to significantly increase frictional stress due to the interplanar-locking effects^{33,49}. Therefore, unlike the conventional ordered PPTs or carbides that contribute to only strengthening, PPTs in the present alloy are able to implement both strengthening and ductilisation mechanisms. Notably, this...

References

- [48.] Choi, W. S. *et al.* Effects of transformation-induced plasticity on the small-scale deformation behavior of single crystalline complex concentrated alloys. *Scr. Mater.* **176**, 122–125 (2020).
- [49.] van Vucht, J. H. N. Influence of radius ratio on the structure of intermetallic compounds of the AB₃ type. *J. Less-Common Met.* **11**, 308–322 (1966).

Question #2-13> As sketched in Fig. 5, the phase transformation plays no role in the grain interiors – surely this is where work hardening is most important? Therefore, is the phase transformation effect of limited impact?

Reply #2-13> We sincerely appreciate the comment pointing out critical points, allowing us to discuss mechanisms more thoroughly. The grain interiors are important for work hardening, and the grain interior precipitations are preferred for effective precipitation hardening because the interior sites where most dislocations glide and interact with obstacles. However, the interior precipitations influence on reducing mean-free paths of dislocations, which in turn leads to the limited work hardening effect. Thus, usual precipitation affects the increment of strength rather than that of work hardening. Regarding the interactions between dislocations and interior hP24 precipitates, please kindly refer to **Reply #1-2**, explaining the massive formation of Orowan bowing loops around the precipitates.

For the alloys in which dislocation glides in grain interiors are inherently difficult, it is necessary to consider the vicinity of grain boundaries for understanding contributions to work hardening. For example, martensitic alloys with fine dislocation cell structures (or substructures) or alloys with massive interior precipitates with small interparticle spacing would likely cause back stress to the newly generated dislocations, leading to premature exhaustion of the dislocation source at grain boundaries. Nevertheless, the presence of boundary phases allowing accommodating plastic deformation by phase transformations would prevent premature exhaustion of dislocation sources at the boundaries from the repeated generation of dislocation and enable additional generation of dislocations. In this respect, the dynamic phase transformation at boundaries has similar effects to TRIP effects in Mn steels or quenching and partitioning (Q&P) steels that consist of metastable austenite at the boundaries. It has been demonstrated that the metastable austenite at the grain boundaries significantly contributes to work hardening indeed through the TRIP effect in those conventional steels. Moreover, a few percent of metastable austenite (1–6%) can still contribute effectively, as observed in several studies of Aermet100, PH13-8 Mo, and Mn steels. Thus, the following sentences have been revised in the discussion part with relevant references added.

- **Discussion.** ... The presence of grain-boundary phases allowing to accommodate plastic deformation by phase transformations would prevent premature exhaustion of dislocation sources at the boundaries through the repeated generation of dislocation.

The current dynamic precipitate transformation at boundaries has similar effects to TRIP effects in Mn steels or quenching and partitioning (Q&P) steels that consist of metastable austenite at the boundaries^{41,42}. It is well known that the dynamic phase transformation of the metastable phase in TRIP steels and alloys postpones plastic instability and thus enhances ductility and work hardening in a large scope^{43,44}. The high fraction of metastable austenite contributes to considerable plasticity; however, it also accompanies a decrease in yield strength, and thus this kind of microstructure cannot implement a class of ultrahigh strength steel. In order to maintain the ultrahigh strength level and gain additional ductility, a fraction of the soft phase should keep a minimum, or its morphology should be tuned to possess high strength level and high mechanical stability. As for the represented case, Aermet100 contains 1–6% metastable austenite, which presents as thin foils with no downside to the strength. Although it is hard to exhibit a significant increment of work-hardening rate or ductility in a large scope such as general high-strength TRIP steels, it can contribute to preventing premature failure and improving toughness effectively, as observed in several studies of Aermet100, PH13-8 Mo, and Mn steels^{40,45,46}. Therefore, it is worth mentioning that the TRIP effect in ultrahigh strength alloys exhibits different performances from the conventional high-strength TRIP steels. In this respect, the present work exploits inherently hard intermetallic phases, which also have no downside to the strength, and their deformable and transformable characteristics provide a two-fold enhancement in strength and ductility via ageing.

References

- [41.] Gao, G. *et al.* Enhanced ductility and toughness in an ultrahigh-strength Mn-Si-Cr-C steel: The great potential of ultrafine filmy retained austenite. *Acta Mater.* **76**, 425–433 (2014).
- [42.] Han, J., Lee, S. J., Jung, J. G. & Lee, Y. K. The effects of the initial martensite microstructure on the microstructure and tensile properties of intercritically annealed Fe-9Mn-0.05C steel. *Acta Mater.* **78**, 369–377 (2014).

Question #2-14> Can the authors comment on whether the precipitates in conventional high-strength maraging alloying also exhibit similar deformation characteristics – e.g., high densities of SFs?

Reply #2-14> Thank the reviewer very much for the interesting question. The precipitates in conventional high-strength maraging alloys mostly contribute to only strengthening as efficient dislocation glide inhibitors through the shearing mechanism or Orowan looping mechanism. Conventional high-strength maraging alloys possess precipitates, namely, NiAl (B2), Ni₃Ti (D0₂₄), and M₂C carbides in the case of secondary hardening steels. The strengthening mechanism depends on interdependent factors; precipitate size, morphology, and crystal structure. For example, the NiAl usually precipitates in a very small size having coherency with the bcc matrix. Owing to the small size (few nanometres) and coherent interfaces, the precipitates are shearable and therefore contribute to strengthening as a shearing mechanism. On the other hand, the Ni₃Ti has a different structure from the bcc matrix, which generally results in a larger size than NiAl and thus forms a semi-coherent interface with the matrix. The semi-coherent nature and the large size make the precipitates difficult to shear for the dislocations acting as barriers, leading to the Orowan looping mechanism. Although numerous studies have been reported regarding dislocation-precipitation interactions, no such case has

been reported where precipitates deform themselves with partial dislocations or stacking faults while strengthening with the Orowan looping mechanism, to the authors' knowledge.

Reviewer #3 (Remarks to the Author):

This manuscript is mainly focusing on the design and development of $\text{Co}_{0.8}\text{FeV}_{0.2}$ medium-entropy alloy with enhanced strength, work-hardening, and ductility, which are dominantly contributed by the formation of semi-coherent precipitates and dynamic precipitate transformation. The Co-Fe-V alloy was well designed by DFT calculation and thoroughly studied by advanced experiment approaches, such as TEM and APT. Overall, I felt that the authors did nice work, and the manuscript is well written. The contents of the manuscript are enough to attract significant attention in HEA society, since the design strategy of precipitate-strengthened MEA/HEA is still not well established, compared to its solid-solution-strengthened MEA/HEA. Thus, I would like to suggest this work be considered for publication after minor revision. I have some minor comments/questions on the manuscript as shown below.

Question #3-1> The authors categorised the studied $\text{Co}_{0.8}\text{FeV}_{0.2}$ alloy as a medium entropy alloy. However, there is no description of the uniqueness of M/HEAs. The M/HEA cannot be only defined by a number of alloy components and their atomic ratio. Please include the definition/description of M/HEA and its benefits to properties (such as entropy effect, lattice distortion, and etc.) in the introduction section.

Reply #3-1> We appreciate the helpful suggestion. We have included the definition and benefits of M/HEAs in the manuscript as follows:

- **Introduction.** ...improve both yield strength and work-hardening behaviour in an initially hard martensitic $\text{FeCo}_{0.8}\text{V}_{0.2}$ medium-entropy alloy (MEA) as a model alloy.

MEAs, as a subclass of alloys termed high-entropy alloys (HEAs), multi-principal element alloys (MPEAs), or compositionally complex alloys (CCAs), consist of generally 3–4 elements at high concentrations, where the high configuration entropy supports the formation of solid-solution phase rather than intermetallic compounds²⁴.

Those alloys exhibit remarkable mechanical properties which originate from high solid-solution strengthening or severe lattice distortion due to large differences in atomic volumes and electronegativity of constituent elements^{25,26}. Based on this strong matrix, precipitates occurring dynamic transformation are selected through Ab-initio calculations that the 50Co–25Fe–25V precipitates show an indistinct difference in the phase stability...

References

- [24.] Yeh, J. W. *et al.* Nanostructured high-entropy alloys with multiple principal elements: Novel alloy design concepts and outcomes. *Adv. Eng. Mater.* **6**, 299–303 (2004).
- [25.] Oh, H. S. *et al.* Engineering atomic-level complexity in high-entropy and complex concentrated alloys. *Nat. Commun.* **10**, 1–8 (2019).
- [26.] Yin, B., Maresca, F. & Curtin, W. A. Vanadium is an optimal element for strengthening in both fcc and bcc high-entropy alloys. *Acta Mater.* **188**, 486–491 (2020).

Question #3-2> Is there any specific reason why the authors have designed this alloy by DFT rather than CALPHAD? If so, please include the advantage of DFT approach for alloy design compared to CALPHAD, since most of HEAs have been designed by either empirical rules and CALPHAD.

Reply #3-2> We appreciate the reviewer for the critical suggestions. First, our primary objective was to design a metastable ordered-structure phase as a precipitate through DFT calculations. We targeted M₃V-type (A₃B-type with M: Fe, Co, Ni) structures from Liu et al. [Ref. 33], as the structures vary from ordered fcc to hcp depending on the electron

concentration (e/a, valence electrons per atom) adjusted by M composition. Ni having the highest number of valence electrons was excluded from the candidate as it can result in high hexagonality and the absence of cubic structure. Therefore, we chose Fe and Co as candidates for the M site to target the metastable intermetallic phase between Co_3V and Fe_3V . Stoichiometric compositions from Co_3V to Fe_3V were considered in the DFT calculation, consequently resulting in the lowest energy gap between L1_2 and hP24 structure at $\text{Co}_2\text{Fe}_1\text{V}_1$. However, the calculated composition represents that of precipitate, which cannot reveal any direct relationship with the designed alloy compositions, as the reviewer commented. In fact, we had designed the alloy composition via CALPHAD approaches via thermodynamic calculations to form the aimed $\text{Co}_2\text{Fe}_1\text{V}_1$ precipitates. At the first submission, we omitted details of thermodynamic calculations to construct concise manuscript structures and also highlight the lowest energy gap. Please kindly refer to **Reply #1-3** regarding details of CALPHAD approaches. The calculation details have been added in the sections Alloy design and Methods, also in the supplementary materials (Supplementary Fig. 2 and 3).

Question #3-3> How the authors obtained the dislocation densities for 3 conditions heat-treated alloys?

Reply #3-3> We appreciate the helpful comment. The dislocation densities had been obtained using a modified Williamson-Hall method based on the measured data from XRD analysis. The obtained dislocation density (ρ) are $1.43 \times 10^{15} \text{ m}^{-2}$ in SA, $9.49 \times 10^{14} \text{ m}^{-2}$ in 1H, and $8.40 \times 10^{14} \text{ m}^{-2}$ in 24H, as provided in the manuscript. The measurement details have been added in Methods as follows.

- **Estimation of strengthening contributions. ... Peak positions and line broadening (represented by the full width at half maximum, FWHM) data of five representative**

peaks of bcc, *i.e.*, (110), (200), (211), (220), and (310), were extracted to estimate dislocation densities of the matrix. We plotted ΔK and $KC^{1/2}$ from the five representative bcc peaks data and fitted linear slope, quantitatively analysing the line broadening. The equation used for fitting the slope is as follows:

$$\Delta K \cong 0.9/D + (\pi\kappa^2b^2/2)^{1/2}\cdot\rho^{1/2}KC^{1/2} + O(K^2C), \quad (3)$$

where θ is the diffraction angle, λ is the wavelength of the X-rays, K is $2\sin\theta/\lambda$, ΔK is $(2\Delta(2\theta)\cos\theta/\lambda)$, D characterises the crystallite size, b is the Burgers vector, C is the average contrast factor of dislocations, and κ is a constant depending on the effective outer cut-off radius of the dislocations. O stands for the higher-order terms where $O(K^2C)$ is considered negligible. ...

Question #3-4> The XRD data should move to the main text.

Reply #3-4> We sincerely thank the reviewer for the helpful suggestion. The XRD data have been moved to Fig. 2a as a part of microstructural characterisations and the following sentences have been added in results section.

Results

- **Microstructure and precipitation behaviour.** Figure 2a exhibits X-ray diffraction (XRD) patterns that only bcc peaks were identifiable in the SA and 1H alloy, while fcc and hcp phases were present in addition to the bcc matrix for 24H alloy. ...

- **Figure 2. Characterisation of precipitates upon ageing conditions.** **a** Phase identification via X-ray diffraction (XRD) analysis for SA, 1H, and 24H alloys. **b–d** Scanning electron microscopy (SEM) images and electron-backscatter diffraction (EBSD) phase maps for different ageing times, **e–i** transmission electron microscopy (TEM) images and atom probe tomography (APT) reconstruction and proximity histogram across precipitates and bcc matrix for 1H alloy. The 50 at% Co iso-concentration surface shows the reference phase boundary. **j,k,m,n** TEM images, **l,o** corresponding fast Fourier-transform (FFT) images, **p,q** APT reconstruction and proximity histogram for 24H alloy.

Question #3-5> The current status of the paper should be re-organized. For instance, some parts of the strengthening and plasticity mechanisms section should be the discussion part and

there is no scientific discussion in the current discussion part. The current discussion part looks like a summary part to me.

Reply #3-5> We sincerely appreciate the suggestion that requires a re-organisation of discussions. We have thoroughly revised the discussion parts including strengthening, ductilisation, and plasticity mechanisms originating from matrix, hP24, and L12 precipitates. The following discussions have been revised in the manuscript, with subheadings to help the reviewer's understanding (not provided in the manuscript).

1. Strengthening mechanism in yield strength increment

- **Discussion.** To elucidate the strengthening and ductilisation mechanisms by maraging effects, the deformed microstructures (tensile strained by 1%) of 24H alloy were investigated through TEM analyses as shown in Fig. 4. It is likely that fine PPTs significantly enhance the yield strength; either by shearing or Orowan bowing mechanism. The dark-field image in Fig. 4a shows the top view of the rod-shaped PPTs, where the direct observation of dislocation-precipitate interactions indicates that gliding dislocations bypass by the Orowan bowing mechanism. The strength increment from the precipitation strengthening of interior hP24 is estimated to be ~1030 MPa (see Methods and Supplementary Fig. 8 for detailed information). On the other hand, the strengthening contribution from forest dislocations decreases from 493 MPa to 377 MPa as the alloy undergoes ageing for 24 h. The reduction in dislocation density is attributed to the combination of thermal recovery (dislocation annihilation and rearrangement) and consumption by forming PPTs with semi-coherent interfaces. As the ageing proceeds for 1 h and 24 h, the dislocation density of the alloy gradually decreases from $1.43 \times 10^{15} \text{ m}^{-2}$ in SA to $9.49 \times 10^{14} \text{ m}^{-2}$ in 1H and $8.40 \times 10^{14} \text{ m}^{-2}$ in 24H. These two different contributions have a counter effect; however, the large increase in

strength due to precipitation renders the decrease in strength due to the reduction in dislocations relatively negligible. However, this precipitation strengthening, on its own, cannot be a unique mechanism for the notable performance of the alloys examined in this study. The aged 24H alloy exhibits greater ductility, specifically uniform elongation, despite its ultrahigh strength level.

2. Mechanisms for the enhanced work hardening and ductilisation I: dislocations behaviour in the matrix

- **Discussion.** The increased strength and ductility of the present alloy are attributed to the following three dominant mechanisms: 1) dislocation behaviours in the matrix; 2) SF formation in interior PPT (hP24); and 3) TRIP effect in grain-boundary PPT (L1₂). First, Fig. 4b,c shows the deformed substructure of the matrix and that adjacent to the interfaces between the interior PPTs and matrix. High-density dislocations form homogeneous deformation substructures as indicated by a blue arrow, and the high fraction and small interspacing of PPTs lead to massive dislocation interactions in the matrix. Before ageing, the as-quenched SA alloy initially possesses high dislocation density due to inherent characteristics of martensitic transformations. This initial high-density tangled dislocation has limited capability of work hardening as represented in Fig. 3a,b. However, the reduction in dislocation density due the ageing treatment (24H alloy) increases the mean free path of dislocations (MFP). This increased MFP allows uniform dislocation glides at a certain regular spacing (see Fig. 4b,c) and consequent high ductility. However, it cannot be concluded that this mechanism is solely dominant in strengthening and ductilisation due to massive interior PPTs with the average interparticle spacing of ~58 nm, which limits the substantial increase of MFP.

Nevertheless, the uniform dislocation glides and homogeneous deformation substructures contribute to preventing premature cracking in ultrahigh-strength alloys.

3. Mechanisms for the enhanced work hardening and ductilisation II: partial dislocations glide in hP24 (the deformable hard precipitate)

- **Discussion.** Secondly, turning the focus to the PPTs, it is observed that the interior hP24 PPTs show the glide of SFs within them as well as the formation of Orowan loops with matrix dislocations. Although the interior PPTs after ageing already contain dense SFs (see Fig. 2k and Supplementary Fig. 5), the FFT image of deformed PPTs shown in Fig. 4d reveals a more evident pattern of the hP24 structure. This result indicates that partial dislocations in the PPTs can glide by applied stress, leading to the well-defined highly faulted structure. Therefore, it is concluded that the interior PPTs accompany dislocation glides, which contributes to plasticity as well as the precipitate strengthening via the Orowan mechanism.

4. Mechanisms for the enhanced work hardening and ductilisation III: dynamic phase transformation of the grain-boundary L12 precipitate

- **Discussion.** Thirdly, whereas the interior PPTs (hP24) possess dense SFs prior to deformation, the grain-boundary PPTs (L1₂) remain as defect-free states (Fig. 2n). Plastic deformation introduces partial dislocations motion and SFs formation within the L1₂ that result in the dynamic phase transformation into hP24, leading to considerable work-hardening and large uniform ductility. It is confirmed that the TRIP occurs after yielding (Fig. 4e,f), and the high critical stress for TRIP can be estimated by an increasing high work hardening rate after yielding in contrast to the SA sample (Fig. 3b)³⁹⁻⁴¹. Figure 4e,f clearly shows the gradual progress of deformation and resulting phase transformation during deformation. Partial dislocations were emitted

from the grain boundary, resulting in the extended SFs and dynamic precipitate transformation into hP24. The presence of grain-boundary phases allowing to accommodate plastic deformation by phase transformations would prevent premature exhaustion of dislocation sources at the boundaries through the repeated generation of dislocation.

The current dynamic precipitate transformation at boundaries has similar effects to TRIP effects in Mn steels or quenching and partitioning (Q&P) steels that consist of metastable austenite at the boundaries^{41,42}. It is well known that the dynamic phase transformation of the metastable phase in TRIP steels and alloys postpones plastic instability and thus enhances ductility and work hardening in a large scope^{43,44}. The high fraction of metastable austenite contributes to considerable plasticity; however, it also accompanies a decrease in yield strength, and thus this kind of microstructure cannot implement a class of ultrahigh strength steel. In order to maintain the ultrahigh strength level and gain additional ductility, a fraction of the soft phase should keep a minimum, or its morphology should be tuned to possess high strength level and high mechanical stability. As for the represented case, Aermet100 contains 1–6% metastable austenite, which presents as thin foils with no downside to the strength. Although it is hard to exhibit a significant increment of work-hardening rate or ductility in a large scope such as general high-strength TRIP steels, it can contribute to preventing premature failure and improving toughness effectively, as observed in several studies of Aermet100, PH13-8 Mo, and Mn steels^{40,45,46}. Therefore, it is worth mentioning that the TRIP effect in ultrahigh strength alloys exhibits different performances from the conventional high-strength TRIP steels. In this respect, the present work exploits inherently hard intermetallic phases, which also have no

downside to the strength, and their deformable and transformable characteristics provide a two-fold enhancement in strength and ductility via ageing.

The underlying mechanism of dynamic structural changes from $L1_2$ to hP24 can be understood based on the stacking faults pair. As observed in $Ni_3(Al,Ti)$ precipitation-hardened nickel-based alloys, $L1_2$ shearing by matrix dislocations can result in different structures of $SnNi_3$ -type $D0_{19}$, $TiNi_3$ -type $D0_{24}$, and VCo_3 -type hP24⁴⁷. The stacking sequence of $L1_2$ is $ABCABCA\dots$, whereas that of hP24 is $ABCACBA\dots$, which exhibits twin-like formation. The stacking sequence of $L1_2$ can be changed to hP24 through the shear displacements of the type $\{111\}1/3\langle 112\rangle$ with superlattice intrinsic (S-ISF) and extrinsic (S-ESF) stacking faults pair⁴⁷. To further illustrate the sequence changes, a schematic drawing of the layers is provided (Supplementary Fig. 9), where the modification of sequences via SFs is shown. Through $\langle 112\rangle$ -type shear displacement of partial dislocations, S-ESF adds a C' layer between A_2 and B_2 , whereas S-ISF removes C_2 layer from the sequence. As a result, the extrinsic/intrinsic stacking faults pair leads to a sequence from $A_1B_1C_1A_2B_2C_2$ to $A_1B_1C_1A_2C'B_2A_3$ as $ABCACBA\dots$, which is that of hP24, *i.e.*, the VCo_3 type.

In this respect, the glide of partial dislocations enables $L1_2$ to hP24 precipitate transformation during deformation, introducing additional phase boundaries and SFs. The introduced interfaces effectively reduce the mean free path of dislocations and even hinder the activation of secondary slip systems due to the limited number of slip systems in the hcp structure^{13,48}. This mechanism, therefore, significantly contributes to work hardening through the dynamic Hall–Petch effect. The phase transformation also relieves the strain energy accumulated during tensile deformation, enabling further plastic deformation to be accommodated. Furthermore, the cubic to hexagonal transformation in the A_3B -type structure is known to significantly increase frictional

stress due to the interplanar-locking effects^{33,49}. Therefore, unlike the conventional ordered PPTs or carbides that contribute to only strengthening, PPTs in the present alloy are able to implement both strengthening and ductilisation mechanisms. Notably, this dynamic phase transformation and consequent ductilisation of ultrastrong alloys are attributed to the narrow stability gap of the desired multiple PPTs. The deformation mechanisms of the PPTs were sketched in Fig. 5 with microstructural evolutions during ageing.

Question #3-6> The target applications for developed M/HEAs are mostly high-temperature. Have authors tried to investigate the mechanical behaviour at elevated temperature for the present alloy?

Reply #3-6> We appreciate very interesting suggestions. Thanks to the suggestion, we have additionally investigated high-temperature mechanical behaviours for the present 24H alloy, specifically at 500 °C at a strain rate of 10^{-3} s^{-1} . The obtained stress-strain curves are as follows.

Figure. Room- and high-temperature engineering stress-strain curves for the 24H alloy.

Note that the slope in the elastic region of high-temperature curve is inaccurate because of the absence of an extensometer in the atmosphere chamber when measuring tensile strains. Although the work-hardening ability seems to deteriorate compared to the room-temperature property, the alloy still shows a very high strength level of approximately 1600 MPa even at 500 °C. We would like to thank the reviewer for the valuable comments on making this interesting approach possible, and we will proceed with further and more diverse follow-up studies based on these results.

REVIEWER COMMENTS

Reviewer #1 (Remarks to the Author):

The authors have satisfactorily addressed all of my issues and I have no further reservations regarding the suitability of the paper for publication.

However, I do have a minor comment for the authors' consideration to improve the quality of the manuscript. The deformation substructure provided in the revised manuscript mainly focuses on the details such as the interaction with nanoprecipitates. It would be nice to provide the information regarding the evolution of basic deformation patterns (such as slip band characteristics) of this specific alloy during the plastic deformation, perhaps at the micrometer scale.

Reviewer #2 (Remarks to the Author):

My comments have been addressed sufficiently. The level of detail and extra work performed by the authors is commendable.

Reviewer #3 (Remarks to the Author):

The authors carefully addressed comments and suggestions. Thus, I strongly suggest this article be published in this journal.

Dear Reviewers,

We would like to submit the following revised manuscript for publication in *Nature Communications*, on behalf of all co-authors.

Title: Doubled strength and ductility via maraging effect and dynamic precipitate transformation in ultrastrong medium-entropy alloy

Authors: Hyun Chung, Won Seok Choi, Hosun Jun, Hyeon-Seok Do, Byeong-Joo Lee, Pyuck-Pa Choi, Heung Nam Han, Won-Seok Ko, Seok Su Sohn*

We cordially thank the reviewers for considering our work and providing helpful and valuable comments. We carefully revised the manuscript accordingly, and a detailed point-by-point reply to each item is enclosed below. Revised or added sentences have been **highlighted in blue** in the revised manuscript. Please kindly consider the attached reply and revised manuscript for further consideration.

Sincerely yours,

Seok Su Sohn

Associate Professor, Korea University

E-mail. sssohn@korea.ac.kr

Reviewer #1 (Remarks to the Author):

The authors have satisfactorily addressed all of my issues and I have no further reservations regarding the suitability of the paper for publication. However, I do have a minor comment for the authors consideration to improve the quality of the manuscript. The deformation substructure provided in the revised manuscript mainly focuses on the details such as the interaction with nanoprecipitates. It would be nice to provide the information regarding the evolution of basic deformation patterns (such as slip band characteristics) of this specific alloy during the plastic deformation, perhaps at the micrometer scale.

Reply #1> We sincerely appreciate the reviewer's helpful comment, allowing us to improve the explanation of deformation evolutions in detail. Although we agree with the reviewer, direct observation and investigation of slip-band characteristics are challenging for such current alloys as they already possess highly tangled and high density of dislocations owing to martensitic microstructure. Besides, unlike the fcc slip systems, evident slip bands are indistinguishable in bcc, where dislocations glide in typical wavy motions dominated by screw dislocations. Nevertheless, as the reviewer recommended, we have put in a great deal of effort to reveal the evolution of deformation structures with respect to dislocation density and deformation band development in SEM and EBSD scales.

EBSD KAM analysis was conducted for the same region before and after tensile deformation (Supplementary Fig. 9a,b). Based on the KAM values, geometrically necessary dislocations (GNDs) were determined using Eq. (1)³²:

$$\rho_{GND} (m^{-2}) = \frac{2\theta}{ub} \quad (1)$$

where u is the unit length, b is the magnitude of Burgers vector, and θ is the misorientation. The estimated dislocation density increases from $2.30 \times 10^{15} m^{-2}$ to $2.70 \times 10^{15} m^{-2}$ during

deformation. Besides, the KAM maps reveal that the dislocation density increases homogeneously throughout the entire grains, where massive dislocation-precipitate interactions occur in nanometre scale as presented in Fig. 4b.

Moreover, evolutions of basic deformation patterns were observed through SEM images in Supplementary Fig. 9c–f. Deformation occurs by forming parallel micro shear bands that pass through the grain interior without crack initiation at the precipitates, indicating the precipitate and matrix well accommodates plastic strain (Supplementary Fig. 9c). At more strained region, more micro shear bands are distributed densely and homogeneously through entire grains (Supplementary Fig. 9d). Then, the homogeneous deformation bands are severely deepened and eventually leads to crack initiation mostly at the prior fcc boundary (Supplementary Fig. 9e). Finally, macro-scale observation on the fractured region exhibit that the matrix sufficiently accommodates deformation bands and thus homogeneous deformation substructures contribute in preventing premature cracking. In other words, the present precipitates can effectively accommodate plastic strains without initiating brittle cracks, which enables a highly sustained strain hardening rate. Thus, the following texts, figures, figure captions, and a reference have been added as follows.

Discussion: ...Nevertheless, the uniform dislocation glides and homogeneous deformation substructures contribute to preventing premature cracking in ultrahigh-strength alloys as observed in Supplementary Fig. 9.

Supplementary References: [32.] Calcagnotto, M., Ponge, D., Demir, E., & Raabe, D. Orientation gradients and geometrically necessary dislocations in ultrafine grained dual-phase steels studied by 2D and 3D EBSD. *Mater. Sci. Eng. A* **527**, 2738–2746 (2010).

Supplementary Fig. 9. Microstructural characterisation of deformation evolutions for 24H alloy. a,b EBSD Kernel average misorientation (KAM) maps of the same area at different strain levels, **a** 0% and **b** 4%. The KAM maps reveal that the dislocation density increases homogeneously throughout the entire grains, where massive dislocation-precipitate interactions occur in nanometre scale as presented in Fig. 4b. **c–f** Observation of deformation bands evolution and failure through SEM images. **c** Parallel micro shear bands formed along prior fcc boundaries and within grains over a wide range, **d** more parallel bands developed uniformly. **e** Severe shear bands lead to crack initiation at the prior fcc boundary. **f** Macro-scale observation on the fractured region with the presence of numerous bands, which indicates uniform and homogeneous deformation occurred.

REVIEWERS' COMMENTS

Reviewer #1 (Remarks to the Author):

The authors have satisfactorily addressed all of my issues and I have no further reservations regarding the suitability of the paper for publication.